# DeepDRK: Deep Dependency Regularized Knockoff for Feature Selection

**Hongyu Shen**[1]    **Yici Yan**[2]    **Zhizhen Zhao**[1]
Department of Electrical and Computer Engineering[1],
University of Illinois at Urbana Champaign.
Department of Statistics[2],
University of Illinois at Urbana Champaign.
{hongyu2, yiciyan2, zhizhenz}@illinois.edu

## Abstract

Model-X knockoff has garnered significant attention among various feature selection methods due to its guarantees for controlling the false discovery rate (FDR). Since its introduction in parametric design, knockoff techniques have evolved to handle arbitrary data distributions using deep learning-based generative models. However, we have observed limitations in the current implementations of the deep Model-X knockoff framework. Notably, the "swap property" that knockoffs require often faces challenges at the sample level, resulting in diminished selection power. To address these issues, we develop "Deep Dependency Regularized Knockoff (DeepDRK)," a distribution-free deep learning method that effectively balances FDR and power. In DeepDRK, we introduce a novel formulation of the knockoff model as a learning problem under multi-source adversarial attacks. By employing an innovative perturbation technique, we achieve lower FDR and higher power. Our model outperforms existing benchmarks across synthetic, semi-synthetic, and real-world datasets, particularly when sample sizes are small and data distributions are non-Gaussian.

## 1 Introduction

Feature selection (FS) has garnered significant attention over the past few decades due to the rapidly increasing dimensionality of data, as well as the associated computational, storage, and noise challenges [22, 15]. Successfully identifying the true informative features among inputs can significantly enhance the performance of analysis frameworks and drive advancements in fields such as biology, neuroscience, medicine, economics, and social sciences [15, 55]. However, the task of accurately pinpointing informative features is often considered nearly impossible, particularly due to the limited availability of data relative to the increasing dimensionality [55, 11]. A practical approach to address this challenge is the development of algorithms designed to select features while maintaining controlled error rates. Targeting this goal, Model-X knockoffs, a novel framework, is proposed in [4, 11] to select relevant features while controlling the false discovery rate (FDR). In contrast to the classical setup, where assumptions on the correlations between input features and the response are imposed [6, 19], the Model-X knockoff framework only requires a linear relationship between the response and the features. With a strong finite-sample FDR guarantee, Model-X knockoff saw broad applications in domains such as biology, neuroscience, and medicine, where the sample size is limited [4, 11, 27, 47, 55, 38, 62].

There have been considerable developments of knockoffs since its debut. In scenarios where feature distributions are complex, various deep learning methods [27, 47, 55, 38, 62] have been proposed. However, we observe major limitations despite improved performance. First, the performance of existing methods varies across different data distributions. Second, the quality of selection declines

when the sample size is relatively small. Third, training the deep knockoff generation models can be challenging due to competing loss terms in the training objective. We elaborate on the drawbacks in Sections 2.2 and 3.2.

In this paper, we address these issues by proposing the Deep Dependency Regularized Knockoff (DeepDRK), a deep learning-based pipeline. We formulate the knockoff generation as an adversarial attack problem involving multiple sources. By optimizing a model against the adversarial environments, we achieve better "swap property" [4] compared to the baseline models. DeepDRK is also equipped with a novel perturbation technique to reduce "reconstructability" [54], which in turn controls FDR and boosts selection power. The experiments conducted on synthetic, semi-synthetic, and real datasets demonstrate that our pipeline outperforms existing methods across various scenarios.

## 2 Background and Related Works

### 2.1 Model-X Knockoffs for FDR control

The Model-X knockoffs framework consists of two main components. Given the explanatory variables $X = (X_1, X_2, \ldots, X_p)^\top \in \mathbb{R}^p$ and the response variable $Y$ ($Y$ continuous for regression and categorical for classification), the framework requires: 1. a knockoff $\tilde{X} = (\tilde{X}_1, \tilde{X}_2, \ldots, \tilde{X}_p)^\top$ that "fakes" $X$; 2. the knockoff statistics $w_j$ for $j \in [p]$ that assess the importance of each feature $X_j$. The knockoff $\tilde{X}$ is required to be independent of $Y$ conditioning on $X$, and must satisfy the swap property:

$$(X, \tilde{X})_{\text{swap}(B)} \stackrel{d}{=} (X, \tilde{X}), \quad \forall B \subset [p]. \tag{1}$$

Here swap$(B)$ exchanges the positions of any variable $X_j$, $j \in B$, with its knockoff $\tilde{X}_j$. The knockoff statistic $w_j((X, \tilde{X}), Y)$ (for $j \in [p]$) depends on the concatenated variable $(X, \tilde{X})$ and $Y$ and must satisfy the flip-sign property:

$$w_j\left((X, \tilde{X})_{\text{swap}(B)}, Y\right) = \begin{cases} w_j((X, \tilde{X}), Y) \text{ if } j \notin B \\ -w_j((X, \tilde{X}), Y) \text{ if } j \in B \end{cases} \tag{2}$$

The functions $w_j(\cdot)$ with $j \in [p]$ have many candidates, for example $w_j = |\beta_j| - |\tilde{\beta}_j|$, where $\beta_j$ and $\tilde{\beta}_j$ are the corresponding regression coefficients of $X_j$ and $\tilde{X}_j$ with the regression function $Y = \sum_{j=1}^p (X_j \beta_j + \tilde{X}_j \tilde{\beta}_j) + \epsilon$, where $\epsilon$ is independently drawn from the normal distribution.

When the two knockoff conditions (i.e., Eq. (1) and (2)) are met, one can select features by $\mathcal{S} = \{w_j \geq \tau_q\}$, where

$$\tau_q = \min_{t>0} \left\{ t : \frac{1 + |\{j : w_j \leq -t\}|}{\max(1, |\{j : w_j \geq t\}|)} \leq q \right\}. \tag{3}$$

To assess the feature selection quality, FDR is commonly used as an average Type I error of selected features [4], which is defined as follows. Let $\mathcal{S} \subset [p]$ be an arbitrary set of selected indices, and let $\beta^*$ denote the underlying true regression coefficients. The FDR for the selection $\mathcal{S}$ is

$$\text{FDR} = \mathbb{E}\left[ \frac{\#\left\{j : \beta_j^* = 0 \text{ and } j \in \mathcal{S}\right\}}{\#\{j : j \in \mathcal{S}\} \vee 1} \right] \tag{4}$$

The control of FDR is guaranteed by the following theorem from [11]:

**Theorem 2.1.** *Given the knockoff that satisfies the swap property in Eq.* (1)*, the knockoff statistic that satisfies Eq.* (2)*, and $\mathcal{S} = \{w_j \geq \tau_q\}$, we have* $\text{FDR} \leq q$.

### 2.2 Related Works

Model-X knockoff is first studied under Gaussian design. Namely, the original variable $X \sim \mathcal{N}(\mu, \Sigma)$ with $\mu$ and $\Sigma$ known. Since Gaussian design does not naturally generalize to complex data distributions, several methods are proposed to weaken the assumption. Among them, model-specific ones

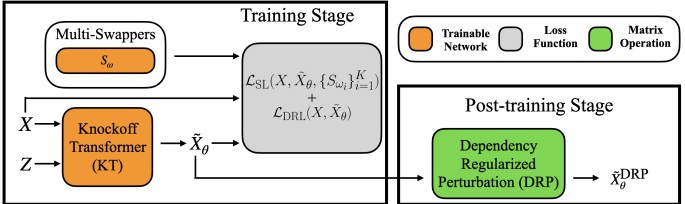

Figure 1: The illustration of the DeepDRK pipeline, which consists of two components: 1. the training stage that optimizes the knockoff Transformer and swappers by $\mathcal{L}_{\text{SL}}$ and $\mathcal{L}_{\text{DRL}}$; 2. the post-training stage that generates the knockoff $\tilde{X}^{\text{DRP}_\theta}$ via dependency regularized perturbation.

such as AEknockoff [34], Hidden Markov Model (HMM) knockoff [51], and MASS [20] all propose parametric alternatives to Gaussian design. These methods can better learn the data distribution, while keeping the sampling process relatively simple. Nevertheless, they pose assumptions to the design distribution, which can be problematic if actual data does not coincide. To gain further flexibility, various deep-learning-based models are developed to generate knockoffs from distributions beyond parametric setup. DDLK [55] and sRMMD [38] utilize different metrics to measure the distances between the original and the knockoff covariates. They apply different regularization terms to impose the "swap property". He et al. [24] introduced a KnockoffScreen procedure to generate multiple knockoffs to improve the stability by minimizing the variance during knockoff construction. KnockoffGAN [27] and Deep Knockoff [47] take advantage of the deep learning structures to create likelihood-free generative models for the knockoff generation.

Despite the flexibility to learn the data distribution, deep-learning-based models suffer from a major drawback. Knockoff generations based on distribution-free sampling methods such as generative adversarial networks (GAN) [21, 3] tend to overfit, namely to learn the data $X$ exactly. The reason is that the notion of swap property for continuous distributions is not well defined at the sample level. To satisfy the swap property, one needs to independently sample $\tilde{X}_j$ from the conditional law $P_{X_j}(\cdot|X_{-j})$, where $X_{-j}$ denotes the vector $(X_1, \ldots, X_{j-1}, X_{j+1}, \ldots, X_p)$. At the sample level, each realization of $X_{-j} = x^i_{-j}$ is almost surely different and only associates to one corresponding sample $X_j = x^i_j$, causing the conditional law to degenerate to sum of Diracs. As a result, minimizing the distance between $(X, \tilde{X})$ and $(X, \tilde{X})_{\text{swap}(B)}$ will push $\tilde{X}$ towards $X$ and introduce high collinearity that makes the feature selection powerless, i.e., with high type II error. To tackle this issue, DDLK [55] suggests an entropic regularization. Yet it still lacks power and is computationally expensive.

### 2.3 Boost Power by reducing reconstructability

The issue of lacking power in the knockoff selection is solved in the Gaussian case [54]. Assuming the knowledge of both mean and covariance of $X \sim \mathcal{N}(\mu, \Sigma)$, the swap property is easily satisfied by setting $\tilde{X}_j \sim \mathcal{N}(\mu_j, \Sigma_{jj})$ and $\Sigma_{ij} = \text{Var}(X_i, \tilde{X}_j)$, for $i \neq j$, $i, j \in [p]$. Barber & Candès [4] originally propose to minimize $\text{Var}(X_j, \tilde{X}_j)$ for all $j \in [p]$ using semi-definite programming (SDP), to prevent $\tilde{X}_j$ to be highly correlated with $X_j$. However, Spector & Janson [54] observed that the SDP knockoff still lacks feature selection power, as merely decorrelating $X_j$ and $\tilde{X}_j$ is not enough, and $(X, \tilde{X})$ can still be (almost) linearly dependent in various cases. This is referred to as high reconstructability [1] in their paper, which can be considered as a population counterpart of collinearity (see Appendix A for more details). To tackle the problem, [54] proposed to maximize the expected conditional variance $\mathbb{E}\text{Var}(X_j \mid X_{-j}, \tilde{X})$, which admits close-form solution whenever $X$ is Gaussian.

## 3 Method

DeepDRK in Figure 1 provides a novel way to generate knockoff $\tilde{X}$ while reducing the reconstructability (see Section 2.3) between the generated knockoff $\tilde{X}$ and the input $X$ for data with

---

[1]The "reconstructability" describes how easy a variable $X_j$ can be constructed deterministically from $X_{-j}$ and $\tilde{X}$ [54].

complex distributions. The generated knockoff can then be used to perform FDR-controlled feature selection following the Model-X knockoff framework (see Section 2.1). Overall, DeepDRK contains two main components. It first trains a transformer-based deep learning model, referred to as Knockoff Transformer (KT), to obtain the swap property and reduce the reconstructability of the generated knockoff. This is achieved by incorporating adversarial attacks with multi-swappers. Secondly, a dependency regularized perturbation technique (DRP) is developed to further reduce the reconstructability for $\tilde{X}$ post training. We will elaborate on these two components in the following two subsections. In this section, we slightly abuse the notation such that $X$ and $\tilde{X}$ also denote the corresponding data matrices.

## 3.1 Training with Knockoff Transformer and Swappers

The KT aims to generate knockoffs, denoted by $\tilde{X}_\theta$, which are parameterized by a transformer network with parameters $\theta$. The loss for training the KT contains a swap loss (SL) $\mathcal{L}_{\text{SL}}$, which enforces the swap property, and a dependency regularization loss (DRL) $\mathcal{L}_{\text{DRL}}$, which controls the reconstructibility of the knockoff. $K$ different neural network parameterized swappers, denoted by $\{S_{\omega_i}\}_{i=1}^K$, are used to test whether the generated knockoffs satisfy the swap property. Thus, the KT is trained adversarially according to the following objective,

$$\min_\theta \max_{\omega_1,\ldots,\omega_K} \left\{ \mathcal{L}_{\text{SL}}(X, \tilde{X}_\theta, \{S_{\omega_i}\}_{i=1}^K) + \mathcal{L}_{\text{DRL}}(X, \tilde{X}_\theta) \right\}. \tag{5}$$

Note that we use $\tilde{X}_\theta$ and $\tilde{X}$ interchangeably; however, the former emphasizes that the knockoff depends on the model weights $\theta$. We discuss each loss function below. Details on the network architectures for KT and swappers are deferred to Appendix B.1. The training algorithm is in Appendix B.2.

### 3.1.1 Swap Loss

The swap loss is designed to enforce the swap property and is defined as follows:

$$\mathcal{L}_{\text{SL}}(X, \tilde{X}_\theta, \{S_{\omega_i}\}_{i=1}^K) = \frac{1}{K} \sum_{i=1}^K \text{SWD}((X, \tilde{X}_\theta), (X, \tilde{X}_\theta)_{S_{\omega_i}})$$
$$+ \lambda_1 \cdot \text{REx}(X, \tilde{X}_\theta, \{S_{\omega_i}\}_{i=1}^K) + \lambda_2 \cdot \mathcal{L}_{\text{swapper}}(\{S_{\omega_i}\}_{i=1}^K), \tag{6}$$

where $\lambda_1$ and $\lambda_2$ are hyperparameters.

The first term in Eq. (6) uses the sliced Wasserstein distance (SWD, see Appendix C for definition) to measure the distance between a pair of joint distributions for $(X, \tilde{X}_\theta)$ and $(X, \tilde{X}_\theta)_{S_{\omega_i}}$, where the swapper is parameterized by $\omega_i$. We sum over $K$ SWDs computed for the distributions modified under different swappers to capture the effects of multiple swap attacks. We utilize SWD to compare distributions because it excels in handling complex data distributions and is computationally efficient [13, 28, 14].

Sudarshan et al. [55] introduced a single swap attack parameterized by a neural network. However, we observe that minimizing the worst case swap for all $B \subset [p]$, as suggested by [55], cannot guarantee the swap property of $\tilde{X}_\theta$. To address this limitation, we introduce a "multi-swapper" setup that uses multiple swappers to enforce the swap property. And this leads to the introduction of the second and the third terms in the objective in Eq. (6).

The second term in Eq. (6), $\text{REx}(X, \tilde{X}_\theta, \{S_{\omega_i}\}_{i=1}^K)$ evaluates the variance of the sliced Wasserstein distances $\text{SWD}((X, \tilde{X}_\theta), (X, \tilde{X}_\theta)_{S_\omega})$ with $K$ realizations of $\omega$ [30]. When $\text{REx}(X, \tilde{X}_\theta, \{S_{\omega_i}\}_{i=1}^K) = 0$, the sliced Wasserstein distances are identical across all swappers. Therefore, complementary to the first term, minimizing this term improves the adherence of swap property for the generated $\tilde{X}_\theta$.

The third term in Eq. (6) is introduced to avoid mode collapse on the parameters $\omega_i$ of different swappers and ensure each swapper characterizes a different adversarial environment:

$$\mathcal{L}_{\text{swapper}}(\{S_{\omega_i}\}_{i=1}^K) = \frac{1}{|C|} \sum_{(i,j)\in C} \text{sim}(S_{\omega_i}, S_{\omega_j}), \tag{7}$$

where $C = \{(i, j)|i, j \in [K], i \neq j\}$, and $\text{sim}(\cdot, \cdot)$ is the cosine similarity between the weights $\omega$ of a pair of different swappers. Without this regularization, all swappers could collapse to a single mode such that the multi-swapper scheme reduces to a single-swapper setup.

Overall, the swap loss $\mathcal{L}_{\text{SL}}$ enforces the swap property via the novel multi-swapper design. Such design provides a more robust assurance of the swap property through multiple adversarial swap attacks, which is shown in the ablation studies in Appendices I.2 and I.3.

### 3.1.2 Dependency Regularization Loss

As discussed in Section 2.2, pursuing the swap property at the sample level often leads to severe overfitting of $\tilde{X}_\theta$, i.e., pushing $\tilde{X}_\theta$ towards $X$, which results in high collinearity in feature selection. To address this, the DRL is introduced to reduce the reconstructability between $X$ and $\tilde{X}$:

$$\mathcal{L}_{\text{DRL}}(X, \tilde{X}_\theta) = \lambda_3 \cdot \text{SWC}(X, \tilde{X}_\theta), \tag{8}$$

where $\lambda_3$ is a hyperparameter. The SWC term in Eq. (8) refers to the sliced Wasserstein correlation [33], which quantitatively measures the dependency between two random vectors in the same space. More specifically, let $Z_1$ and $Z_2$ be two $p$-dimensional random vectors. $\text{SWC}(Z_1, Z_2) = 0$ indicates that $Z_1$ and $Z_2$ are independent, while $\text{SWC}(Z_1, Z_2) = 1$ suggests a linear relationship between each other (see Appendix D for more details on SWC). In DeepDRK, we minimize SWC to reduce the reconstructability, a procedure similar to [54]. The intuition is as follows. If the joint distribution of $X$ is known, then for each $j \in [p]$, the knockoff $\tilde{X}_j$ should be sampled from $P_j(\cdot|X_{-j})$, making $X_j$ and $\tilde{X}_j$ less dependent. In such case the swap property is ensured, and collinearity/reconstructability is reduced due to independence. As we do not have access to the joint law, we want the variables to be less dependent. Since collinearity exists with in $X$, merely decorrelate $X_j$ and $\tilde{X}_j$ is not enough. Thus, we minimize SWC to reduce the dependence between $X$ and $\tilde{X}$. We refer readers to Appendix A for more discussions.

## 3.2 Dependency Regularization Perturbation

Empirically we observe a competition between $\mathcal{L}_{\text{SL}}$ and $\mathcal{L}_{\text{DRL}}$ in Eq. (5), which adds difficulty to the training procedure. Specifically, the $\mathcal{L}_{\text{SL}}$ is dominating and the $\mathcal{L}_{\text{DRL}}$ increases quickly after a short decreasing period. We are the first to observe this phenomenon in all deep-learning based knockoff generation models when one tries to gain power [47, 55, 38, 27]. We include the experimental evidence in Appendix E. We suggest the following explanation: minimizing the swap loss, which corresponds to FDR control, is the same as controlling Type I error. Similarly, minimizing the dependency loss is to control Type II error. With a fixed number of observations, it is well known that Type I error and Type II error can not decrease at the same time after reaching a certain threshold. In the framework of model-X knockoff, we aim to boost as much power as possible given the FDR is controlled at a certain level, a similar idea as the uniformly most powerful (UMP) test [12]. For this reason, we propose DRP as a post-training technique to further boost power.

DRP is a sample-level perturbation that further eliminates dependency between $X$ and and the knockoff. More specifically, DRP perturbs the generated $\tilde{X}_\theta$ with the row-permuted version of $X$, denoted as $X_{\text{rp}}$. After applying DRP, the final knockoff $\tilde{X}_{\theta,n}^{\text{DRP}}$ becomes:

$$\tilde{X}_{\theta,n}^{\text{DRP}} = (1 - \alpha_n) \cdot \tilde{X}_\theta + \alpha_n \cdot X_{\text{rp}}, \tag{9}$$

where $\alpha_n$ is a preset perturbation weight, $n$ is the sample size, and $\alpha_n \to 0$ when $n \to \infty$. In the following, we remove $n$ or $\theta$ whenever the context is clear. $\tilde{X}^{\text{DRP}}$ has a smaller SWC with $X$, since $X_{\text{rp}}$ is independent of $X$. Despite the perturbation increases the swap loss, such impact is negligible when the sample size is large. More specifically, we present the following Lemma 3.1 and Proposition 3.2. Let $\hat{P}_n$ and $\hat{P}_n^B$ denote the empirical joint distribution of the sample $X$ and $\tilde{X}$ before and after the swap: $(X, \tilde{X}) \sim \hat{P}_n$, $(X, \tilde{X})_{\text{swap}(B)} \sim \hat{P}_n^B$. And $P$ and $P^B$ denote their corresponding population distributions.

**Lemma 3.1.** *Under mild conditions[41], the slice Wasserstein distance between the empirical distributions of $(X, \tilde{X})$ and $(X, \tilde{X})_{\text{swap}(B)}$ and their corresponding population distributions is of scale $\mathcal{O}(n^{-1/2})$, i.e., $SWD(\hat{P}_n, \hat{P}_n^B) = SWD(P, P^B) + \mathcal{O}(n^{-1/2})$.*

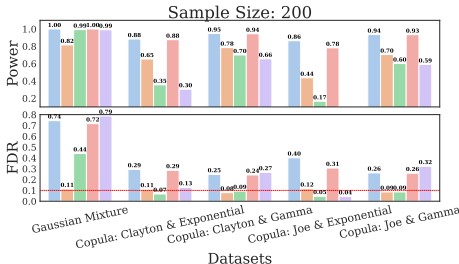
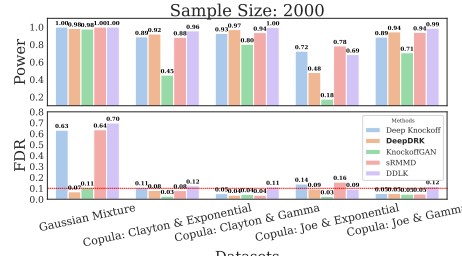

Figure 2: Power and FDR for different knockoff models on the synthetic datasets with $\beta \sim \frac{p}{15 \cdot \sqrt{N}} \cdot$ Rademacher(0.5). The red horizontal line indicates the 0.1 FDR threshold.

**Proposition 3.2.** *Let* $\alpha_n \lesssim \mathcal{O}(n^{-1/2})$ *in Eq.* (9)*, and denote* $(X, \tilde{X}^{DRP}) \sim \hat{P}_{n,DRP}$, $(X, \tilde{X}^{DRP})_{\text{swap}(B)} \sim \hat{P}^B_{n,DRP}$. *Then* $SWD(\hat{P}_{n,DRP}, \hat{P}^B_{n,DRP}) = SWD(P, P^B)$ *when* $n \to \infty$.

The proofs can be found in Appendix F. Lemma 3.1 suggests a general case where the swap loss enforced at the sample level differs from that of the population level from a term of scale $n^{-1/2}$, which vanishes when $n \to \infty$. Proposition 3.2 provides the rationale on the proposal of the perturbation technique in Eq. (9), such that the DRP term is negligible asymptotically. Besides the theoretical justification, we also empirically show in Appendix G that the perturbation is beneficial.

# 4 Experiment

We assess the performance of DeepDRK against several benchmark models under three experimental setups: 1. a fully synthetic setup where both the input variables and the response variable follow predefined distributions; 2. a semi-synthetic setup, in which the input variables are derived from real-world datasets and the response variable is generated based on known relationships with the inputs; and 3. feature selection (FS) using a real-world dataset. The experiments are designed to encompass a range of datasets with varying $p/n$ ratios and distributions of $X$, aiming to provide a comprehensive evaluation of model performance. The benchmark models are Deep Knockoff [47] [2], DDLK [55] [3], KnockoffGAN [27] [4] and sRMMD [38] [5], with links of the code implementation listed in the footnote. The implementation details for the training, feature selection, and data preparation are available in Appendix H. In the following, we describe the datasets and the associated experimental results. We also consider the ablation study to illustrate the benefits obtained by having the following proposed terms: REx, $\mathcal{L}_{\text{swapper}}$, the multi-swapper setup, and the dependency regularization perturbation. Empirically, these terms help to improve power and control the FDR. Due to space limitation, we defer details for the ablation studies to Appendix I.2 and I.3. DeepDRK is implemented in PyTorch [45] and is accessible at: `https://github.com/nowonder2000/DeepDRK`.

## 4.1 The Synthetic Experiments

To properly evaluate the performance, we follow a well-designed experimental setup by [55, 38] to generate different datasets specified by $(X, Y)$. Here $X \in \mathbb{R}^p$ is the collection dependent variables that follows pre-defined distributions. $Y \in \mathbb{R}$ is the response variable that is modeled as $Y \sim \mathcal{N}(X^T\beta, 1)$. The underlying true $\beta$ is a $p$-dimensional vector, where each entry is drawn independently from the distribution $\frac{p}{15 \cdot \sqrt{n}} \cdot$ Rademacher(0.5).

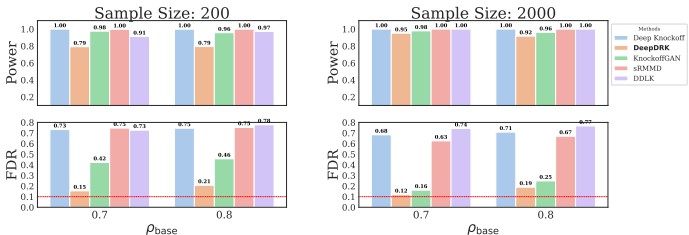

Figure 3: Power and FDR for different knockoff models on the mixture of Gaussian data on different $\rho_{\text{base}}$ setups. The red horizontal line indicates the 0.1 FDR threshold. This figure is complementary to Figure 2 for including two additional Gaussian mixture data with higher $\rho_{\text{base}}$ values.

---

[2]`https://github.com/msesia/deepknockoffs`
[3]`https://github.com/rajesh-lab/ddlk`
[4]`https://github.com/vanderschaarlab/mlforhealthlabpub/tree/main/alg/knockoffgan`
[5]`https://github.com/ShoaibBinMasud/soft-rank-energy-and-applications`

| $n$ | Method | FDR | | | | | Power | | | | |
|---|---|---|---|---|---|---|---|---|---|---|---|
| | | mean | std | median | 5% quantile | 95% quantile | mean | std | median | 5% quantile | 95% quantile |
| | DDLK | 0.772 | 0.025 | 0.781 | 0.726 | 0.791 | 0.971 | 0.033 | 0.983 | 0.911 | 0.995 |
| | Deep Knockoff | 0.735 | 0.028 | 0.740 | 0.692 | 0.769 | 0.999 | 0.001 | 0.999 | 0.997 | 1.000 |
| 200 | KnockoffGAN | 0.390 | 0.110 | 0.396 | 0.230 | 0.550 | 0.971 | 0.038 | 0.986 | 0.904 | 0.997 |
| | sRMMD | 0.720 | 0.047 | 0.741 | 0.640 | 0.752 | 0.998 | 0.002 | 0.998 | 0.994 | 1.000 |
| | **DeepDRK** | 0.116 | 0.040 | 0.100 | 0.086 | 0.187 | 0.791 | 0.039 | 0.804 | 0.720 | 0.824 |
| | DDLK | 0.725 | 0.050 | 0.734 | 0.667 | 0.778 | 1.000 | 0.000 | 1.000 | 1.000 | 1.000 |
| | Deep Knockoff | 0.626 | 0.092 | 0.672 | 0.460 | 0.694 | 1.000 | 0.001 | 1.000 | 0.997 | 1.000 |
| 2000 | KnockoffGAN | 0.118 | 0.014 | 0.114 | 0.102 | 0.139 | 0.973 | 0.012 | 0.978 | 0.953 | 0.986 |
| | sRMMD | 0.658 | 0.060 | 0.644 | 0.583 | 0.736 | 1.000 | 0.001 | 1.000 | 0.998 | 1.000 |
| | **DeepDRK** | 0.081 | 0.018 | 0.076 | 0.059 | 0.110 | 0.973 | 0.011 | 0.978 | 0.956 | 0.983 |

Table 1: Comparison of different methods on FDR and power across different $(\pi_1, \pi_2, \pi_3)$ for the components in the Gaussian mixture setup.

Compared to the previous works [55, 38], which consider $\frac{p}{\sqrt{n}}$ as the scaling factor for the Rademacher distribution, we reduce the magnitude of $\beta$ by a factor of 15. This is because we find that in the original setup, the $\beta$ scale is too large such that the feature selection enjoys high powers and low FDRs for all models. To compare the performance of the knockoff generation methods on various data, we consider the following distributions for $X$:

`Gaussian mixture`: We consider a Gaussian mixture model $X \sim \sum_{k=1}^3 \pi_k \mathcal{N}(\mu_k, \Sigma_k)$, where $\pi$ is the proportion of the $k$-th Gaussian with $(\pi_1, \pi_2, \pi_3) = (0.4, 0.2, 0.4)$. $\mu_k \in \mathbb{R}^p$ denotes the mean of the $k$-th Gaussian with $\mu_k = \mathbf{1}_p \cdot 20 \cdot (k-1)$, where $\mathbf{1}_p$ is the $p$-dimensional vector that has universal value 1 for all entries. $\Sigma_k \in \mathbb{R}^{p \times p}$ is the covariance matrices whose $(i, j)$-th entry taking the value $\rho_k^{|i-j|}$, where $\rho_k = \rho_{\text{base}}^{k-0.1}$ and $\rho_{\text{base}} = 0.6$. Besides this experiment, we further perform two additional tests. The first one focuses on a mixture of Gaussians data of various $\rho_{\text{base}}$ to study the feature selection performance with highly correlated features. Namely, we consider additional $\rho_{\text{base}} \in \{0.7, 0.8\}$. The second one explores the effect of $(\pi_1, \pi_2, \pi_3)$ to the feature selection performance. In this case, we uniformly draw 10 sets of $(\pi_1, \pi_2, \pi_3)$, and evaluate the FS performance of all the models considered in this paper. The values of the mixture weights are presented in Table 4 in Appendix H.3.1.

`Copulas` [49]: We further use copula to model complex correlations within $X$. To the best of our knowledge, this is a first attempt to consider complex distributions other than the Gaussian mixture model in the knockoff framework. Specifically, we consider two copula families: Clayton, Joe with the consistent copula parameter of 2 in both cases. For each family, we consider two candidates for the marginal distributions: a uniform distribution (using the identity conversion function) and an exponential distribution with a rate of 1. We implement the copulas according to `PyCop` [6].

We consider the following $(n, p)$ setups: $(200, 100)$ and $(2000, 100)$. This is in contrast to existing works, which consider only the $(2000, 100)$ as the smallest sample size setup [55, 38, 47]. Our goal is to demonstrate the consistent performance of DeepDRK across various $p/n$ ratios, particularly when the sample size is small.

`Results`: Figure 2 compare FDRs and powers for all models across the datasets with two different setups for $\beta$. Figure 2 shows that DeepDRK consistently controls the false discovery rate (FDR) compared to other benchmark models across various data distributions and $p/n$ ratios, with the exception of a few cases where the FDR exceeds the 0.1 threshold in the small sample size ($n = 200$) scenarios. Other models, though being able to reach higher power, comes at a cost of sacrificing FDR, which contradicts to the UMP philosophy (see Section 3.2). We also evaluate the performance on a mixture of Gaussians with increasing $\rho_{\text{base}}$, indicating a higher correlation among the input variables. Note that the mixture of Gaussians example in Figure 2 has $\rho_{\text{base}} = 0.6$. The results for $\rho_{\text{base}} \in \{0.7, 0.8\}$ are presented in Figure 3. Compared to other baseline models, DeepDRK maintains the lowest FDRs while achieving competitive powers across all $\rho_{\text{base}}$ values, highlighting its robustness to correlations in $X$. Overall, the results demonstrate the ability of DeepDRK in consistently performing FS with controlled FDR compared to other models across a range of different datasets, $p/n$ ratios, and feature correlations in $X$. In addition, we present common statistics for the FDR and power on the mixture of Gaussians experiment with 10 different sets of $(\pi_1, \pi_2, \pi_3)$ in Table 1. It is clear that DeepDRK improves the robustness in maintaining FDR across various configurations of the Gaussian mixture models compared to existing approaches.

---

[6] https://github.com/maximenc/pycop/

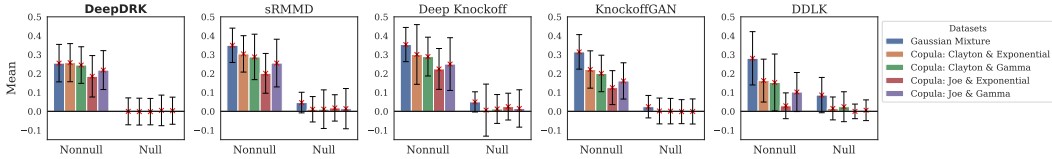

Figure 4: The knockoff statistics ($w_j$) for different knockoff models on the synthetic datasets with $\beta \sim \frac{p}{15 \cdot \sqrt{N}} \cdot \text{Rademacher}(0.5)$. Each bar in the plot represents the mean of the null/nonnull knockoff statistics averaging on 600 experiments. The error bar indicates the standard deviation. The sample size is 200.

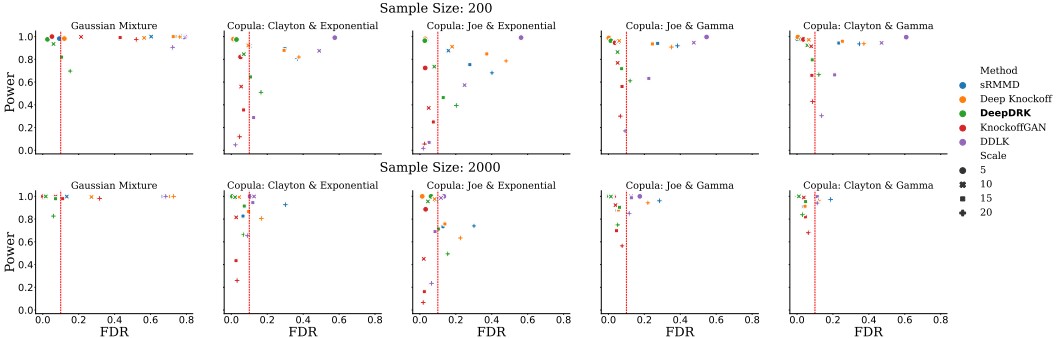

Figure 5: Scatter plots of Power against FDR for different datasets and models. The red vertical line indicates the 0.1 FDR threshold. Different scales for $\beta$ (e.g., $\frac{p}{5 \cdot \sqrt{n}}$, $\frac{p}{10 \cdot \sqrt{n}}$, $\frac{p}{15 \cdot \sqrt{n}}$ and $\frac{p}{20 \cdot \sqrt{n}}$) are indicated by different marker styles. Different models are indicated by different colors.

To understand why DeepDRK outperforms other baseline models, we consider measuring the distribution of the knockoff statistics, i.e., $w_j$, for both nonnull and null features of $X$. Fan et al. [17] and Candès et al. [11] pointed out that a good knockoff requires the corresponding knockoff statistics to concentrate symmetrically around zero for the null features and to maintain high positive values for the nonnulls. However, theoretical analysis on the goodness of FDR or power requires access to the true knockoff $\tilde{X}$ [17] to compare the distribution of $w_j$'s with the ground truth, which is infeasible for non-Gaussian data. Nevertheless, we can still examine the distribution of the knockoff statistics as a surrogate to analyze model performance in terms of false discovery rate (FDR) or power, given the necessary properties of the knockoff statistics mentioned earlier.

Figure 4 shows the means and standard deviations of the empirical distributions of the knockoff statistics $w_j$ for both null and nonnull variables across different datasets and models. Clearly, compared to other benchmarks, DeepDRK maintains high values of $w_j$ for the nonnulls and relatively symmetric values around zero for the nulls. All other models experience positive shifts to the null statistics to some extent. Positive shifts in the null statistics lead to a degeneracy in performance because the threshold selection rule for false discovery rate (FDR) is based on the negative values of $w_j$'s (see Eq. (3)). This has two negative impacts, one on the FS threshold and the other on its subsequent FS process. First, according to Eq. (3), the shift causes the chosen threshold to approach zero, as there are fewer null statistics remaining on the negative side, and those that do remain have smaller amplitudes. Subsequently, a lowered threshold leads to an increase in false positives, a phenomenon that becomes more pronounced with positive shifts in the null statistics. As shown in Figure 4, the $w_j$ values calculated using DeepDRK are centered around zero for the nulls, while exhibiting large positive values for the nonnulls. This aligns with the results in Figure 2, which demonstrate that DeepDRK effectively achieves good FDR control and high power. Due to space limit, we defer the results for $n = 2000$ and the comparison of $w_j$ statistics for the Gaussian correlation setup to Appendix I.3.

To further verify the performance consistency of the proposed method, we include a comparison on different $\beta$ scales. Specifically, we consider 4 different sets of $\beta$ scales, i.e., $\frac{p}{5 \cdot \sqrt{n}}$, $\frac{p}{10 \cdot \sqrt{n}}$, $\frac{p}{15 \cdot \sqrt{n}}$ and $\frac{p}{20 \cdot \sqrt{n}}$, for the previously considered data distributions. The results are summarized in Figure 5. It is observed that in all cases considered with various $\beta$ scale, the proposed DeepDRK successfully

maintains relatively low FDRs with a higher power, compared to baseline methods. This phenomenon is especially pronounced with a low sample size ($n = 200$).

In addition to the above results, we provide the measurement of the swap property in Appendix I.1. The evaluation of model runtime is also included in Appendix I.4.

## 4.2 The Semi-synthetic Experiments

We consider a semi-synthetic study with design $X$ drawn from two real-world datasets and use $X$ to simulate response $Y$. The first dataset contains single-cell RNA sequencing (scRNA-seq) data from $10\times$ Genomics [7]. Each entry in $X \in \mathbb{R}^{n \times p}$ represents the observed gene expression of $p$ genes in $n$ cells. We refer readers to [23] and [1] for more background. Following the same preprocessing in [23], we obtain the final dataset $X$ with $n = 10000$ and $p = 100$ [8]. The preprocessing of $X$ and the synthesis of $Y$ (denoted as "Linear" and "Tanh" cases) are deferred to Appendix H.3.2.

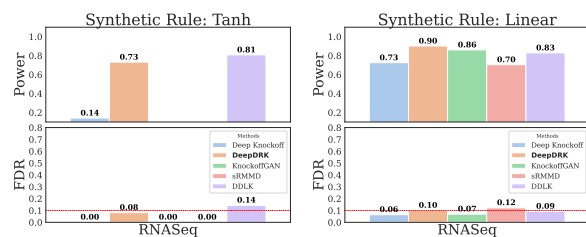

Figure 6: Power and FDR for different knockoff models on the semi-synthetic RNA dataset. The red horizontal line indicates the 0.1 FDR threshold.

The second publicly available dataset[9] is from a real case study entitled "Longitudinal Metabolomics of the Human Microbiome in Inflammatory Bowel Disease (IBD)" [35]. The study seeks to identify important metabolites of two representative diseases of the inflammatory bowel disease (IBD): ulcerative colitis (UC) and Crohn's disease (CD). Specifically, we use the C18 Reverse-Phase Negative Mode dataset that has 546 samples and 91 metabolites. To mitigate the effects of missing values, we preprocess the dataset following a common procedure to remove metabolites that have over 20% missing values, resulting in 80 metabolites. We normalize the data matrix entry-wise to have zero mean and unit variance after a log transform and an imputation via the $k$-nearest neighbor algorithm following the same procedure in [38]. Finally, we synthesize the response $Y$ with the real dataset of $X$ via $Y \sim \mathcal{N}(X^T\beta, 1)$, where the entries of $\beta$ drawn independently from one of the following three distributions: Unif$(0, 1)$, $\mathcal{N}(0, 1)$, and Rademacher$(0.5)$, in three separate experiments.

`Results`: Figure 6 and 7 compare the feature selection performance on the RNA data and the IBD data respectively. In Figure 6, we observe that all but DDLK are bounded by the nominal 0.1 FDR threshold in the "Tanh" case. However, KnockoffGAN and sRMMD have almost zero power. The power for Deep Knockoff is also very low compared to that of DeepDRK. Although DDLK provides high power, the associated FDR is not controlled by the threshold. In the "Linear" case, almost all models have well controlled FDR, among which DeepDRK provides the highest power. Similar observations can be found in Figure 7. For the IBD data generated under the aforementioned synthesis rules, it is clear that all models except DDLK achieve well-controlled FDR. Apart from DDLK, DeepDRK consistently demonstrates the highest power. These results further underscore the potential of DeepDRK for real-world data applications.

## 4.3 A Case Study

Besides (semi-)synthetic setups, we carry out a case study with real data for both design $X$ and response $Y$, in order to qualitatively evaluate the selection performance of DeepDRK. In this subsection, we use the IBD dataset [35] with the empirical response. The response variable $Y$ is categorical: $Y$ equals 1 if a given sample is associated with UC/CD and 0 otherwise. The covariates $X$ is identical to the second semi-synthetic setup considered in Section 4.2. To properly evaluate results with no ground truth available, we search for the evidence of the IBD-associated metabolites using the existing literature. Specifically, we use three sources: 1. metabolites that are explicitly

---

[7]`https://kb.10xgenomics.com/hc/en-us`

[8]The data processing code is adopted from this repo: `https://github.com/dereklhansen/flowselect/tree/master/data`

[9]`https://www.metabolomicsworkbench.org/` under the project DOI: 10.21228/M82T15.

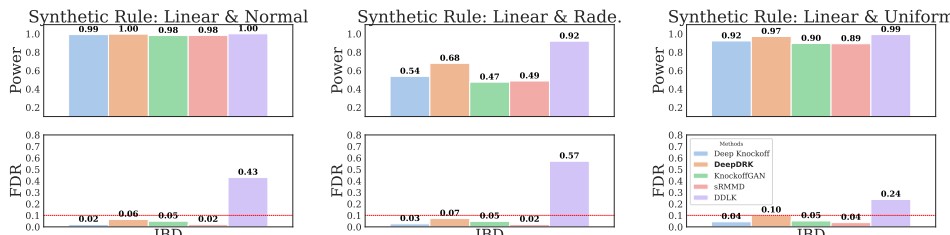

Figure 7: Power and FDR for different knockoff models on the semi-synthetic IBD dataset. The red horizontal line indicates the 0.1 FDR threshold.

| Model | DeepDRK | Deep Knockoff | sRMMD | KnockoffGAN | DDLK |
|---|---|---|---|---|---|
| Referenced / Identified | 19/23 | 15/20 | 5/5 | 12/14 | 17/25 |

Table 2: The number of literature-supported metabolites among the identified metabolites vs. the number of identified metabolites.

documented to have associations with IBD, UC, or CD in the PubChem database [10]; 2. metabolites that are reported in the existing peer-reviewed publications; 3. metabolites that are reported in pre-prints. We identify 47 metabolites that are reported to have association with IBD. All referenced metabolites are included in Table 5 in Appendix H.3.3.

Our DeepDRK model training and knockoff generation are the same as before (see Table 3 in Appendix H.1). Likewise, to generate knockoff for the benchmark models, we follow their default setups. During the FS step, however, we use 0.2 as the FDR threshold instead of 0.1, and apply a different algorithm—DeepPINK [37] that is included in the `knockpy` [11] library—to generate $w_j$. The values are subsequently used to identify metabolites. We choose DeepPINK over the previously considered ridge regression due to the nonlinear relationships between metabolites $X$ and responses $Y$ in this case study.

We compare the FS results with the 47 literature-supported metabolites and report the number of selections in Table 2. A detailed list of selected features for each model can be found in Table 9 in Appendix I.5. From Table 2, it is clear that, compared to the benchmark models, DeepDRK identifies the largest number of referenced metabolites while effectively limiting the number of metabolites not reported in existing literature (see Table 5 in Appendix H.3.3). The sRMMD model achieves the lowest false discovery rate, but this comes at the cost of missing a significant number of documented metabolites. Since there is no ground truth available, the results here should be viewed and analyzed qualitatively.

## 5 Conclusion

In this paper, we introduce DeepDRK, a deep learning-based knockoff generation pipeline consisting of two steps. First, it trains a Knockoff Transformer with multiple swappers to achieve the swap property while reducing reconstructability. In the post-training stage, a dependency-regularized perturbation is applied to further enhance power with controlled FDR. DeepDRK effectively balances FDR and power, which compete with each other at the sample level. To the best of our knowledge, this relationship has not been previously reported in the literature. Empirically, DeepDRK demonstrates the ability to maintain both controlled false discovery rates (FDR) and high power across various data distributions and different $p/n$ ratios. Additionally, we provide insights into the distribution of knockoff statistics, which elucidate the reasons behind DeepDRK's consistently strong performance. The numerical results indicate that DeepDRK outperforms existing deep learning-based benchmark models. Experiments with real and semi-synthetic data further highlight the potential of DeepDRK for feature selection tasks involving non-Gaussian data.

---

[10] https://pubchem.ncbi.nlm.nih.gov/

[11] https://amspector100.github.io/knockpy/

## Acknowledgments

This research was partially supported by Alfred P. Sloan foundation and NSF #1934757.

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

# Appendix

## A   Reconstructability and Selection Power

The concept of reconstructability was introduced by [54] as a population-level counterpart to what is commonly referred to as collinearity in linear regression. Under a Gaussian design where $X \sim \mathcal{N}(0, \Sigma)$, reconstructability is high if $\Sigma$ is not of full rank, implying that there exists some $j$ such that $X_j$ is almost surely a linear combination of $X_{-j}$. More generally, if there exists more than one representation of the response $Y$ using the explanatory variable $X$, we qualitatively consider the reconstructability to be high. High collinearity often impairs statistical power, and similarly, high reconstructability diminishes feature selection power. To illustrate this, we state a linear version of Theorem 2.3 from [54], originally formulated for a more general single-index model [25].

**Theorem A.1.** *Let $Y = X_J \beta_J + X_{-J} \beta_{-J} + \varepsilon$, where $J \subset [p]$ and $\varepsilon$ is centered Gaussian noise. Equivalently, this implies $Y \perp\!\!\!\perp X_J \mid X_J \beta_J, X_{-J}$. Suppose there exists a $\beta_J^*$ such that $X_J \beta_J = X_J \beta_J^*$ almost surely. Then, denoting $Y^* = X_J \beta_J^* + X_{-J} \beta_{-J} + \varepsilon$, we have*

$$((X, \tilde{X}), Y) \stackrel{d}{=} \left( (X, \tilde{X})_{\mathrm{swap}(J)}, Y^* \right) \quad and$$

$$\left( (X, \tilde{X}), Y^* \right) \stackrel{d}{=} \left( (X, \tilde{X})_{\mathrm{swap}(J)}, Y \right). \tag{10}$$

*Furthermore, in the knockoff framework [4], let $w = w((X, \tilde{X}), y)$ and $w^* = w((X, \tilde{X}), y^*)$. Then, for all $j \in J$,*

$$\mathbb{P}\left( w_j > 0 \right) + \mathbb{P}\left( w_j^* > 0 \right) \leqslant 1. \tag{11}$$

Eq. (11) implies a "no free lunch" situation for selection power when there is exact reconstructability.

To address the reconstructability issue, [54] proposed two methods in Gaussian design. The first is the minimal variance-based reconstructability (MVR) knockoff, in which the knockoff $\tilde{X}$ is sampled to minimize the loss

$$L_{\mathrm{MVR}} = \sum_{j=1}^{p} \frac{1}{\mathbb{E}\left[ \mathrm{Var}\left( X_j \mid X_{-j}, \tilde{X} \right) \right]}. \tag{12}$$

Note that this is equivalent to maximizing $\mathbb{E}\left[ \mathrm{Var}\left( X_j \mid X_{-j}, \tilde{X} \right) \right]$ for all $j \in [p]$. Another approach is the maximum entropy (ME) knockoff, where $\tilde{X}$ is sampled to maximize

$$L_{\mathrm{ME}} = \int \int p(x, \tilde{x}) \log(p(x, \tilde{x})) \, d\tilde{x} \, dx. \tag{13}$$

Under Gaussian design, both optimizations have closed-form solutions. Since $X$ is Gaussian, $(X, \tilde{X})$ must also be jointly Gaussian to satisfy the swap property. To optimize, one first calculates the covariance matrix using the SDP method in [4], yielding a diagonal matrix $S$. Then, both MVR and ME reduce to an optimization on $S$:

$$L_{\mathrm{MVR}}(S) \propto \mathrm{Tr}\left( G_S^{-1} \right) = \sum_{j=1}^{2p} \frac{1}{\lambda_j(G_S)}$$

$$\text{and} \quad L_{\mathrm{ME}}(S) = \log \det \left( G_S^{-1} \right) = \sum_{j=1}^{2p} \log\left( \frac{1}{\lambda_j(G_S)} \right). \tag{14}$$

Although both methods show high power for feature selection, neither MVR nor ME can be directly extended to arbitrary distributions due to the intractability of conditional variance and likelihood.

In DeepDRK, we consider regularizing with a sliced-Wasserstein-based dependency correlation, which can be considered a stronger dependency regularization than entropy. A post-training perturbation is also applied to further reduce collinearity. However, the theoretical understanding of how these affect the swap property and power remains an open question.

# B DeepDRK Model Architecture and Training Algorithm

## B.1 Model Architecture

DeepDRK's knockoff Transformer (KT) model is based on the popular Vision Transformer (ViT) [16]. The primary difference is that the input dimension is 1D, not 2D, for $X$. We do not use patches as input; instead, we treat all entries of $X$ to account for correlations between each pair of entries in the knockoff $\tilde{X}$. This structure is similar to the original Transformer [59]. Nonetheless, we retain other components from ViT, such as patch embedding, PreNorm, and 1D-positional encoding [16]. Since knockoff generation requires a distribution, we feed $X$ and a uniformly distributed random variable $Z$ with the same dimension as $X$ to inject randomness. Specifically for the DeepDRK model, we use 8 attention heads, 8 layers, and a hidden dimension of 512 in the ViT model.

The swapper module, first introduced in DDLK [55], produces the index subset $B$ for the adversarial swap attack. Optimizing knockoff against these adversarial swaps enforces the swap property. Specifically, the swapper consists of a matrix with shape $2 \times p$ (i.e., trainable model weights), where $p$ is the dimension of $X$. This matrix controls the Gumbel-softmax distribution [26] for all $p$ entries. Each entry is represented by a binary Gumbel-softmax random variable (i.e., it can only take values of 0 or 1). To generate the subset $B$, we sample $b_j$ from the corresponding $j$-th Gumbel-softmax random variable, and $B$ is defined as $\{j \in [p] \; ; \; b_j = 1\}$. During optimization, we maximize Eq. (5) with respect to the weights $\omega_i$ of the swapper $S_{\omega_i}$, so that the sampled indices, with which the swap is applied, lead to a higher SWD in the objective in Eq. (5). Minimizing this objective with respect to $\tilde{X}_\theta$ requires the knockoff to counteract the adversarial swaps, thereby enforcing the swap property. Compared to DDLK, the proposed DeepDRK utilizes multiple independent swappers (i.e., $K = 2$). And we set the temperature for the Gumbel-softmax to be 0.2.

## B.2 Training Algorithm

In Algorithm 1, we provide pseudo code for training the Knockoff Transformer and the swappers (i.e., the first stage shown in Figure 1).

---

**Algorithm 1** DeepDRK Training

---

1: **Input:** Knockoff transformer $\tilde{X}_\theta$, denoted as $g_\theta(\cdot)$; swappers $S_\omega$; number of swappers $K$; learning rate $\alpha_s$ for the swappers; learning rate $\alpha_\theta$ for the knockoff transformer; early stop tolerance $\eta$; number of epochs $T$; batch size $B_s$; dataset $\mathcal{D}$; swapper update frequency $\gamma = 3$
2: **Output:** $\theta$ for $\tilde{X}_\theta$
3: Split dataset $\mathcal{D}$ into training set $\mathcal{D}_{\text{train}}$ and validation set $\mathcal{D}_{\text{val}}$
4: Initialize the knockoff transformer $g_\theta(\cdot)$ with random weights
5: Initialize swappers $S_{\omega_i}, i = 1, \ldots, K$ with random weights
6: Initialize the AdamW optimizer $\text{opt}_\theta$ with learning rate $\alpha_\theta$ for $g_\theta(\cdot)$
7: Initialize the AdamW optimizer $\text{opt}_{\omega_i}$ with learning rate $\alpha_s$ for $S_{\omega_i}, i = 1, \ldots, K$
8: **for** $t = 1$ **to** $T$ **do**
9:     **for** $l = 1$ **to** $\frac{|\mathcal{D}_{\text{train}}|}{B_s}$ **do**
10:         Sample $B_s$ samples of $X$ from $\mathcal{D}_{\text{train}}$: $X_l$
11:         Generate knockoff $\tilde{X}_l = g_\theta(X_l)$
12:         Calculate $\mathcal{L}_{\text{SL}}(X_l, \tilde{X}_l, \{S_{\omega_i}\}_{i=1}^K)$ and $\mathcal{L}_{\text{DRL}}(X_l, \tilde{X}_l)$
13:         $\theta \leftarrow \theta + \text{opt}_\theta(\mathcal{L}_{\text{SL}}(X_l, \tilde{X}_l, \{S_{\omega_i}\}_{i=1}^K) + \mathcal{L}_{\text{DRL}}(X_l, \tilde{X}_l))$
14:         **if** $l \bmod \gamma = 0$ **then**
15:             $\omega_i \leftarrow \omega_i + \text{opt}_{\omega_i}(-\mathcal{L}_{\text{SL}}(X_l, \tilde{X}_l, \{S_{\omega_i}\}_{i=1}^K)), i = 1, \ldots, K$
16:         **end if**
17:     **end for**
18:     Calculate the validation loss on $\mathcal{D}_{\text{val}}$: $\mathcal{L}_{\text{SL}}^{\text{val}} + \mathcal{L}_{\text{DRL}}^{\text{val}}$
19:     **if** $\mathcal{L}_{\text{SL}}^{\text{val}} + \mathcal{L}_{\text{DRL}}^{\text{val}}$ meets early stop condition at tolerance $\eta$ **then**
20:         **break**
21:     **end if**
22: **end for**

---

## C From Wasserstein to Sliced Wasserstein Distance

The Wasserstein distance has gained popularity in both mathematics and machine learning due to its ability to compare different types of distributions [60] and its differentiability [3]. Here, we provide its definition. Let $X, Y$ be two $\mathbb{R}^d$ random vectors following distributions $P_X, P_Y$ with finite $p$-th moments. The *Wasserstein-$p$* distance between $P_X$ and $P_Y$ is:

$$W_p(P_X, P_Y) = \inf_{\gamma \in \Gamma(P_X, P_Y)} \left( \mathbb{E}_{(x,y) \sim \gamma} \|x - y\|^p \right)^{\frac{1}{p}} \tag{15}$$

where $\Gamma(P_X, P_Y)$ denotes the set of all joint distributions such that their marginals are $P_X$ and $P_Y$. When $d = 1$, the Wasserstein distance between two one-dimensional distributions can be written as:

$$W_p(P_X, P_Y) = \left( \int_0^1 |F_X^{-1}(v) - F_Y^{-1}(v)|^p dv \right)^{\frac{1}{p}} \tag{16}$$

$$= \|F_X^{-1} - F_Y^{-1}\|_{L^p([0,1])},$$

where $F_X$ and $F_Y$ are the cumulative distribution functions (CDFs) of $P_X$ and $P_Y$, respectively. Moreover, if $p = 1$, the Wasserstein distance further simplifies to

$$W_1(P_X, P_Y) = \int |F_X(v) - F_Y(v)| dv = \|F_X - F_Y\|_{L^1(\mathbb{R})}. \tag{17}$$

From the above, it is evident that the 1D Wasserstein distance is straightforward to compute, leading to the development of the sliced Wasserstein distance (SWD) [9]. To leverage this computational advantage in 1D, one first projects both distributions uniformly onto a 1D direction and computes the Wasserstein-$p$ distance between the two projected distributions. SWD is then calculated by taking the expectation over the random direction. Specifically, let $\mu \in \mathbb{S}^{d-1}$ denote a projection direction, and the push-forward distribution [60] $\mu_\sharp P_X$ denotes the law of $\mu^T X$. When $\mu$ is uniformly distributed on the $d$-dimensional sphere, the $p$-sliced Wasserstein distance between $P_X$ and $P_Y$ is given by:

$$SW_p(P_X, P_Y) = \int_{\mu \in \mathbb{S}^{d-1}} W_p(\mu_\sharp P_X, \mu_\sharp P_Y) \, d\mu. \tag{18}$$

Combining Eq. (18) and Eq. (16) yields:

$$SW_p(P_X, P_Y) =$$

$$\int_{\mu \in \mathbb{S}^{d-1}} \left( \int_0^1 |(F_X^\mu)^{-1}(v) - (F_Y^\mu)^{-1}(v)|^p dv \right)^{\frac{1}{p}} d\mu \tag{19}$$

$$= \int_{\mu \in \mathbb{S}^{d-1}} \int |F_X^\mu(v) - F_Y^\mu(v)| dv \, d\mu \quad \text{when } p = 1. \tag{20}$$

Despite faster computation, the convergence of SWD has been shown to be equivalent to the convergence of the Wasserstein distance under mild conditions [10]. In practice, the expectation over $\mu$ is approximated by a finite summation over a number of projection directions chosen uniformly from $\mathbb{S}^{d-1}$.

## D Sliced Wasserstein Correlation

The idea of metricizing independence is recently advanced using the Wasserstein distance [61, 43]. Given a joint distribution $(X, Y) \sim \Gamma_{XY}$ and its marginal distributions $X \sim P_X, Y \sim P_Y$, the Wasserstein Dependency (WD) between $X$ and $Y$ is defined by $\text{WD}(X, Y) = W_p(\Gamma_{XY}, P_X \otimes P_Y)$. A trivial observation is that $\text{WD}(X, Y) = 0$ implies that $X$ and $Y$ are independent. Due to the high computational cost of Wasserstein distance, sliced Wasserstein dependency (SWDep) [33] is developed using sliced Wasserstein distance (see Appendix C for SWD details). The SW dependency (SWDep) between $X$ and $Y$ is defined as $\text{SW}_p(\Gamma_{XY}, P_X \otimes P_Y)$, and a zero SWDep indicates independence. Since the dependency metric is not bounded from above, sliced Wasserstein correlation

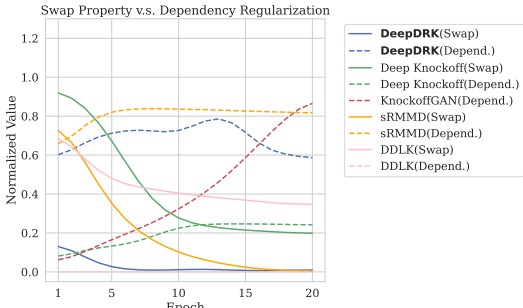

Figure 8: The competing relationship between the swap property ($\mathcal{L}_{\mathrm{SL}}$ in solid curves) and dependency regularization ($\mathcal{L}_{\mathrm{DRL}}$ in dashed curves).

(SWC) is introduced to normalize SWDep. More specifically, the SWC between $X$ and $Y$ is defined as

$$\mathrm{SWC}_p(X,Y) := \frac{\mathrm{SWDep}_p(X,Y)}{\sqrt{\mathrm{SWDep}_p(X,X)\,\mathrm{SWDep}_p(Y,Y)}}$$
$$= \frac{\mathrm{SWD}_p(\Gamma_X Y, P_X \otimes P_Y)}{\sqrt{\mathrm{SWD}_p\left(\Gamma_{XX}, P_X \otimes P_X\right)\mathrm{SWD}_p\left(\Gamma_{YY}, P_Y \otimes P_Y\right)}}, \tag{21}$$

where $\Gamma_X X$ and $\Gamma_Y Y$ are the joint distributions of $(X,X)$ and $(Y,Y)$ respectively. It is shown that $0 \leqslant \mathrm{SWC}_p(X,Y) \leqslant 1$ and $\mathrm{SWC}_p(X,Y) = 1$ when $X$ has a linear relationship with $Y$ [33].

In terms of computing SWC, we follow [33] and consider both laws of $X$ and $Y$ to be sums of $2n$ Diracs, i.e., both variables are empirical distributions with data $\mathcal{I}_{\mathrm{full}} = \{(\mathbf{x}_i, \mathbf{y}_i)\}_{i=1}^{2n}$. Define $\mathcal{I} = \{(\mathbf{x}_i, \mathbf{y}_i)\}_{i=1}^{n}$ and $\tilde{\mathcal{I}} = \{(\tilde{\mathbf{x}}_i, \tilde{\mathbf{y}}_i)\}_{i=1}^{n}$, where $(\tilde{\mathbf{x}}_i, \tilde{\mathbf{y}}_i) = (\mathbf{x}_{n+i}, \mathbf{y}_{n+i})$. Because data is i.i.d., $\mathcal{I}$ and $\tilde{\mathcal{I}}$ are independent. We further introduce the following notation:

$$\mathcal{I}_{\mathbf{xy}} = \{(\mathbf{x}_i, \mathbf{y}_i)\}_{i=1}^{n}, \tilde{\mathcal{I}}_{\mathbf{xy}} = \{(\tilde{\mathbf{x}}_i, \mathbf{y}_i)\}_{i=1}^{n}$$
$$\mathcal{I}_{\mathbf{xx}} = \{(\mathbf{x}_i, \mathbf{x}_i)\}_{i=1}^{n}, \tilde{\mathcal{I}}_{\mathbf{xx}} = \{(\tilde{\mathbf{x}}_i, \mathbf{x}_i)\}_{i=1}^{n},$$
$$\mathcal{I}_{\mathbf{yy}} = \{(\mathbf{y}_i, \mathbf{y}_i)\}_{i=1}^{n}, \tilde{\mathcal{I}}_{\mathbf{yy}} = \{(\tilde{\mathbf{y}}_i, \mathbf{y}_i)\}_{i=1}^{n}$$

Then the empirical SWC can be computed by:

$$\widehat{\mathrm{SWC}}_p(X,Y) := \frac{\mathrm{SWD}_p\left(I_{\mathcal{I}_{\mathbf{xy}}}, I_{\tilde{\mathcal{I}}_{\mathbf{xy}}}\right)}{\sqrt{\mathrm{SWD}_p\left(I_{\mathcal{I}_{\mathbf{xx}}}, I_{\tilde{\mathcal{I}}_{\mathbf{xx}}}\right)\mathrm{SWD}_p\left(I_{\mathcal{I}_{\mathbf{yy}}}, I_{\tilde{\mathcal{I}}_{\mathbf{yy}}}\right)}}. \tag{22}$$

## E Competing Losses

In Figure 8, we present scaled $\mathcal{L}_{\mathrm{SL}}$ and $\mathcal{L}_{\mathrm{DRL}}$ curves for each model considered. For better visualization, these curves are normalized to values between 0 and 1. We observe the first 20 epochs, as $\mathcal{L}_{\mathrm{DRL}}$ flattens out in later epochs without further decrease. The competition between losses is evident: as $\mathcal{L}_{\mathrm{SL}}$ decreases, $\mathcal{L}_{\mathrm{DRL}}$ increases, indicating difficulty in maintaining low reconstructability.

## F Proofs

### F.1 Proof of Lemma 3.1

Assuming all conditions in [41] hold, we prove this by applying inequalities on $\mathrm{SWD}(\hat{P}_n, \hat{P}_n^B)$ and $\mathrm{SWD}(P, P^B)$:

$$\mathrm{SWD}(\hat{P}_n, \hat{P}_n^B) \leqslant \mathrm{SWD}(\hat{P}_n, P) + \mathrm{SWD}(P, P^B) + \mathrm{SWD}(P^B, \hat{P}_n^B)$$
$$= \mathrm{SWD}(P, P^B) + \mathcal{O}(n^{-1/2}), \tag{23}$$

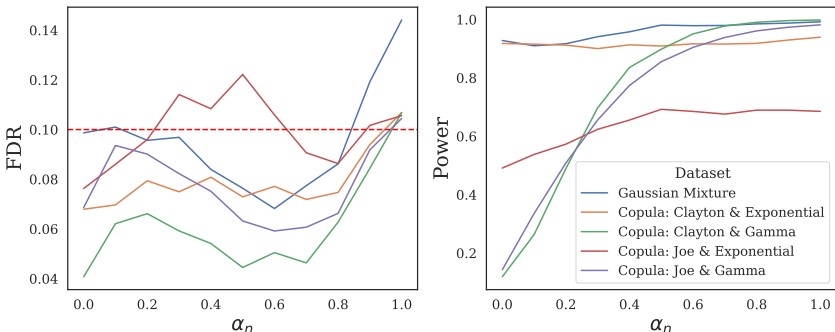

Figure 9: The effect of $\alpha_n$ for $\tilde{X}_\theta^{\text{DRP}}$ in Eq. (8) on FDR and power. When $\alpha_n$ is 0, we consider the knockoff generated from the knockoff transformer without any dependency regularization perturbation. When $\alpha_n$ is 1, there is only perturbation $X_{\text{rp}}$ without the knockoff. The sample size $n = 2000$.

where the first inequality is a direct application of the triangle inequality, and the second equality follows from Theorem 5 in [41]. Similarly, we obtain

$$\text{SWD}(P, P^B) \leqslant \text{SWD}(P, \hat{P}_n) + \text{SWD}(\hat{P}_n, \hat{P}_n^B) + \text{SWD}(\hat{P}_n^B, P^B)$$
$$= \text{SWD}(\hat{P}_n, \hat{P}_n^B) + \mathcal{O}(n^{-1/2}). \tag{24}$$

Combining Eq. (23) and Eq. (24), we have

$$\text{SWD}(P, P^B) = \text{SWD}(\hat{P}_n, \hat{P}_n^B) + \mathcal{O}(n^{-1/2}). \tag{25}$$

### F.2 Proof of Proposition 3.2

From Eq. (9), we know $\tilde{X}_{\theta,n}^{\text{DRP}} \xrightarrow{a.s.} \tilde{X}_\theta$ as $n \to \infty$. Combining this with Eq. (25), we obtain $\text{SWD}(P, P^B) = \text{SWD}(\hat{P}_n, \hat{P}_n^B) = \text{SWD}(\hat{P}_{n,\text{DRP}}, \hat{P}_{n,\text{DRP}}^B)$ as $n \to \infty$.

# G  Effect of $\alpha_n$ in $\tilde{X}_\theta^{\text{DRP}}$

As discussed in Section 3.2, obtaining the swap property while maintaining low reconstructability at the sample level is empirically challenging. To address this, dependency regularization perturbation (DRP) is introduced. In this section, we evaluate the effect of $\alpha_n$ in $\tilde{X}_\theta^{\text{DRP}}$ in Eq. (9) on feature selection performance. Results are summarized in Figure 9 for five synthetic datasets with the sample size $n = 2000$.

When $\alpha_n$ is decreased, we observe an increase in power. However, the FDR exhibits a bowl-shaped pattern, consistent with the findings of [54]: introducing the permuted $X_{\text{rp}}$ reduces reconstructability, thereby increasing power. However, an overly dominant $X_{\text{rp}}$ compromises the swap property, resulting in higher FDRs. Based on our hyperparameter search and the results in Figure 9, we recommend choosing $\alpha_n$ within the range of 0.4 to 0.5.

## H  Implementation Details

### H.1  Training

To fit models, we first split datasets of $X$ into training and validation sets with an 8:2 ratio. The training sets are used for model optimization, and the validation sets are used for early stopping based on the validation loss, with a patience period of 6. Since knockoffs are not unique [11], testing sets are not required. To evaluate DeepDRK's performance, we compare it with four state-of-the-art (SOTA) deep-learning-based knockoff generation models, focusing on non-parametric data. Specifically, we

| Parameter | Value | Parameter | Value |
|---|---|---|---|
| $S_\omega$ Learning Rate | $1 \times 10^{-3}$ | $\tilde{X}_\theta$ Learning Rate | $1 \times 10^{-5}$ |
| Dropout Rate | 0.1 | # of Epochs | 200 |
| Batch Size | 64 | $\lambda_1$ | 30.0 |
| $\lambda_2$ | 1.0 | $\lambda_3$ | 20.0 |
| Early Stop Tolerance | 6 | $\alpha_n$ | 0.5 |

Table 3: Training configuration.

include Deep Knockoff [47][12], DDLK [55][13], KnockoffGAN [27][14], and sRMMD [38][15]. We use the recommended hyperparameter settings for each model, with the only difference being the number of training epochs, set to 200 for consistency across models.

We follow the model configuration in Table 3 to optimize DeepDRK. The architecture of the swappers $S_\omega$ is based on [55]. Both the swappers and $\tilde{X}_\theta$ are trained using the AdamW optimizer [36]. During training, we alternately optimize $\tilde{X}_\theta$ and the swappers $S_\omega$, updating weights $\theta$ three times for each update of weights $\omega$. This training scheme is similar to GAN training [21], though without discriminators. We apply early stopping to prevent overfitting. A pseudocode of the optimization is provided in Appendix B.2. In experiments, we set $\alpha_n = 0.5$ universally as the dependency regularization coefficient due to its consistent performance [16]. A discussion on the effect of $\alpha_n$ is provided in Appendix G.

Once trained, models generate the knockoff $\tilde{X}$ given $X$ data. The generated $\tilde{X}$ is then combined with $X$ for feature selection. Experiments are conducted on a single NVIDIA V100 16GB GPU.

## H.2 Feature Selection

Once $\tilde{X}$ is obtained, feature selection is performed in three steps. We first concatenate $X$ and $\tilde{X}$ to form an $n \times 2p$ design matrix $(X, \tilde{X})$. In the second step, we use Ridge regression to get the estimated regression coefficients $\{\hat{\beta}_j\}_{j=1}^{2p}$ from $Y$ and $(X, \tilde{X})$. Finally, we compute the knockoff statistics $w_j = |\hat{\beta}_j| - |\hat{\beta}_{j+p}|$, for $j = 1, 2, \ldots, p$, and then select features using the threshold defined in Eq. (3). We set $q = 0.1$ as the FDR threshold, following its common use in other knockoff-based feature selection studies [47, 38]. Each experiment is repeated 600 times, with results reported as the average power and FDR over these 600 repetitions.

## H.3 Data Preparation

### H.3.1 Weights for the Gaussian mixture models

Table 4 includes 10 sets of $(\pi_1, \pi_2, \pi_3)$ for the Gaussian mixture models.

### H.3.2 Preparation of the RNA Data

We first normalize the raw data $X$ to value range $[0, 1]$ and then standardize it to have zero mean and unit variance. $Y$ is synthesized according to $X$. We consider two different ways of synthesizing $Y$. The first, denoted as "Linear", is similar to the previous setup in the full synthetic case with $Y \sim \mathcal{N}(X^T\beta, 1)$ and $\beta \sim \frac{p}{12.5 \cdot \sqrt{N}} \cdot$ Rademacher(0.5). For the second, denoted as "Tanh", the response $Y$ is

| Set No. | $\pi_1$ | $\pi_2$ | $\pi_3$ |
|---|---|---|---|
| 1 | 0.562 | 0.384 | 0.054 |
| 2 | 0.430 | 0.168 | 0.402 |
| 3 | 0.317 | 0.324 | 0.359 |
| 4 | 0.316 | 0.388 | 0.296 |
| 5 | 0.439 | 0.488 | 0.073 |
| 6 | 0.314 | 0.041 | 0.645 |
| 7 | 0.656 | 0.282 | 0.062 |
| 8 | 0.200 | 0.300 | 0.500 |
| 9 | 0.500 | 0.300 | 0.200 |
| 10 | 0.333 | 0.333 | 0.333 |

Table 4: 10 sets of $(\pi_1, \pi_2, \pi_3)$ for the Gaussian mixture models.

---

[12]https://github.com/msesia/deepknockoffs

[13]https://github.com/rajesh-lab/ddlk

[14]https://github.com/vanderschaarlab/mlforhealthlabpub/tree/main/alg/knockoffgan

[15]https://github.com/ShoaibBinMasud/soft-rank-energy-and-applications

[16]We present this value for its consistent performance; however, it may not be the globally optimal value. Future work may improve on $\alpha_n$'s design.

| Reference Type | Metabolite | Source | Meatbolite | Source |
|---|---|---|---|---|
| PubChem | palmitate | CID: 985 | taurocholate | CID: 6675 |
| | cholate | CID: 221493 | p-hydroxyphenylacetate | CID: 127 |
| | linoleate | CID: 5280450 | deoxycholate | CID: 222528 |
| | taurochenodeoxycholate | CID: 387316 | | |
| Publications | 12.13-diHOME | [7] | dodecanedioate | [7] |
| | arachidonate | [7] | eicosatrienoate | [7, 5] |
| | eicosadienoate | [7] | docosapentaenoate | [7, 5] |
| | taurolithocholate | [7] | salicylate | [7] |
| | saccharin | [7] | 1.2.3.4-tetrahydro-beta-carboline-1.3-dicarboxylate | [7] |
| | oleate | [5] | arachidate | [5] |
| | glycocholate | [5] | chenodeoxycholate | [5] |
| | phenyllactate | [38, 31] | glycolithocholate | [5] |
| | urobilin | [38, 46] | caproate | [38, 32] |
| | hydrocinnamate | [38, 29] | myristate | [38, 18] |
| | adrenate | [38, 35] | olmesartan | [38, 48] |
| | tetradecanedioate | [56, 39] | hexadecanedioate | [56, 39] |
| | oxypurinol | [8] | porphobilinogen | [40] |
| | caprate | [52, 53] | undecanedionate | [32, 58] |
| | stearate | [2, 5] | oleanate | [44] |
| | glycochenodeoxycholate | [50] | sebacate | [32] |
| | nervonic acid | [57] | lithocholate | [5] |
| Preprints | alpha-muricholate | [42] | tauro-alpha-muricholate/tauro-beta-muricholate | [42] |
| | 17-methylstearate | [42] | myristoleate | [42] |
| | taurodeoxycholate | [42] | ketodeoxycholate | [42] |

Table 5: IBD-associated metabolites that are supported by the literature. This table includes all 47 referenced metabolites for the IBD case study. Each metabolite is supported by one of the three sources: PubChem, peer-reviewed publications or preprints. For PubChem case, we report the PubChem reference ID (CID), and for the other two cases we report the publication references.

generated following the expression:

$$k \in [m/4]$$
$$\varphi_k^{(1)}, \varphi_k^{(2)} \sim \mathcal{N}(1, 1)$$
$$\varphi_k^{(3)}, \varphi_k^{(4)}, \varphi_k^{(5)} \sim \mathcal{N}(2, 1)$$
$$Y \mid X = \epsilon + \sum_{k=1}^{m/4} \varphi_k^{(1)} X_{4k-3} + \varphi_k^{(3)} X_{4k-2}$$
$$+ \varphi_k^{(4)} \tanh \left( \varphi_k^{(2)} X_{4k-1} + \varphi_k^{(5)} X_{4k} \right), \tag{26}$$

where $\epsilon$ follows the standard normal distribution and the 20 covariates are sampled uniformly.

### H.3.3 Metabolite Information for the IBD Study

Here we provide the implementation details for the experiments described in Section 4.3. In Table 5, we provide all 47 referenced metabolites based on our comprehensive literature review.

## I Additional Results

In this section, we include all results that are deferred from the main paper.

### I.1 Swap Property

We use different metrics to empirically evaluate the swap property on the generated knockoff $\tilde{X}$ and the original data $X$ according to Eq. (1). In this paper, three metrics are considered: mean discrepancy distance with linear kernel, or "MMD(Linear)" for

| Abbreviation | Full Name |
|---|---|
| J+G | Copula: Joe & Gamma |
| C+G | Copula: Clayton & Gamma |
| C+E | Copula: Clayton & Exponential |
| J+E | Copula: Joe & Exponential |
| MG | Mixture of Gaussians |

Table 6: The abbreviations of the names for the datasets.

| | Dataset | MMD(Linear) | $SWD_1$ | $SWD_2$ |
|---|---|---|---|---|
| DDLK | J+G | 2.68 | 0.18 | 0.08 |
| | C+G | 2.49 | 0.18 | 0.07 |
| | C+E | 1.80 | 0.15 | 0.05 |
| | J+E | 2.29 | 0.15 | 0.06 |
| | MG | 306.49 | 2.15 | 9.99 |
| KnockoffGAN | J+G | 7.01 | 0.15 | 0.08 |
| | C+G | 5.11 | 0.16 | 0.04 |
| | C+E | 0.52 | 0.06 | 0.01 |
| | J+E | 1.24 | 0.08 | 0.02 |
| | MG | 1.08 | 3.28 | 26.09 |
| Deep Knockoff | J+G | 13.09 | 0.21 | 0.11 |
| | C+G | 19.04 | 0.27 | 0.14 |
| | C+E | 6.65 | 0.18 | 0.07 |
| | J+E | 6.78 | 0.19 | 0.13 |
| | MG | 2770.00 | 8.98 | 196.54 |
| **DeepDRK (Ours)** | J+G | 0.47 | 0.13 | 0.06 |
| | C+G | 0.71 | 0.14 | 0.05 |
| | C+E | 0.23 | 0.09 | 0.02 |
| | J+E | 0.14 | 0.10 | 0.04 |
| | MG | 380 | 6.13 | 84.37 |
| sRMMD | J+G | 130.02 | 0.68 | 0.71 |
| | C+G | 175.74 | 0.73 | 0.96 |
| | C+E | 49.09 | 0.41 | 0.35 |
| | J+E | 33.24 | 0.35 | 0.29 |
| | MG | 3040 | 8.51 | 142.99 |

Table 7: Evaluation of the swap property. This table empirically measures the swap property by three different metrics: MMD(Linear), $SWD_1$, and $SWD_2$. The evaluation considers all baseline models and all datasets in the synthetic dataset setup. For space consideration, we use abbreviations to indicate the name of the datasets. The full name can be found in Table 6.

short; sliced Wasserstein-1 distance ($SWD_1$); and sliced Wasserstein-2 distance ($SWD_2$). We measure the sample level distances between the vector that concatenates $X$ and $\tilde{X}$ (i.e., $(X, \tilde{X})$) and the vector after randomly swapping the entries (i.e., $(X, \tilde{X})_{\text{swap}(B)}$). To avoid repetition, please refer to section 2.1 and Eq. (1) for the definition of notation. Empirically, it is time consuming to evaluate all subsets $B$ of the index set $[p]$. As a result, we alternatively define a swap ratio $r_s \in \{0.1, 0.3, 0.5, 0.7, 0.9\}$. The swap ratio controls the amount of uniformly sampled indices (i.e., the cardinality $|B| = r_s \cdot p$) in a subset of $[p]$. For any $X$ and $\tilde{X}$ from the same experiment, 5 different subsets $B$ are formed according to 5 different swap ratios. We report the average values over the swap ratios to represent the empirical quantification of the swap property. Results can be found in Table 7.

Clearly, compared to other models, the proposed DeepDRK achieves the smallest values in almost every entry across the three metrics and the first 4 datasets (i.e., J+G, C+G, C+E and J+E in Table 7). This explains why DeepDRK has lower FDRs as DeepDRK maintains the swap property relatively better than the baseline models (see results in Figure 2). Similarly, we observe that KnockoffGAN also achieves relatively small values, which leads to well-controlled FDRs compared to other baseline models. Overall, this verifies the argument in Candès et al. [11] that the swap property is important in guaranteeing FDR during feature selection.

However, we observe a difference for Gaussian mixture data. The proposed DeepDRK achieves the best performance in FDR control and power (see results in Figure 2), yet its swap property measured under the metrics in Table 7 is not the lowest. Despite this counter-intuitive observation, we want to highlight that it does not conflict with the argument in Candès et al. [11]. Rather, it supports our statement that the low reconstructability and the swap property cannot be achieved at the sample level (e.g., the free lunch dilemma in practice). After all, the swap property is not the only factor that determines FDR and power during feature selection.

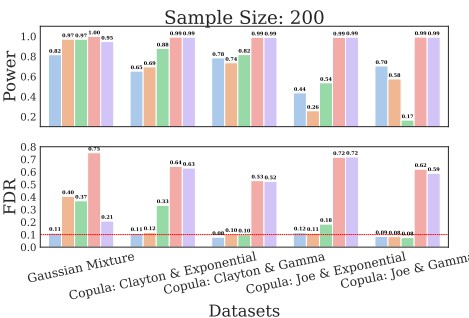
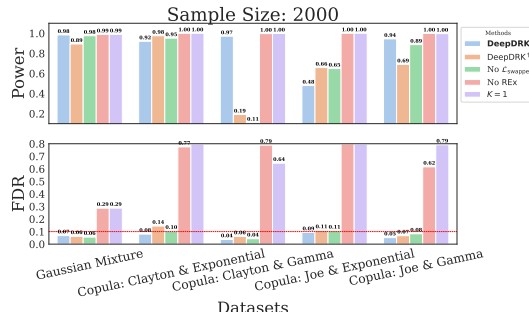

Figure 10: Power and FDR in the ablation studies. The red horizontal line indicates the 0.1 FDR threshold. DeepDRK[†] is the model with dependency regularized perturbation removed. $K$ indicates the number of swappers.

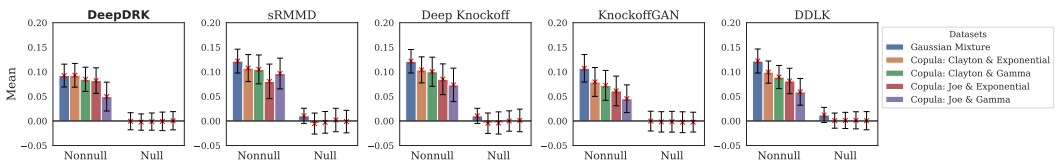

Figure 11: Knockoff statistics ($w_j$) for different knockoff models on the synthetic datasets. Each bar in the plot represents the mean of the null/nonnull knockoff statistics averaging on 600 experiments. The error bar indicates the standard deviation. The sample size is 2000.

## I.2 Ablation Studies

In this subsection, we perform ablation studies on different terms introduced in Section 3.1.1, to show the necessity of designing these terms during the optimization for knockoffs. We consider the fully synthetic setup described in Section 4.1, and consider $n = 200$ and $n = 2000$. The distribution of $\beta$ is $\frac{p}{15 \cdot \sqrt{N}} \cdot \text{Rademacher}(0.5)$. We test the following terms: 1. REx; 2. the number of swappers $K$; 3. $\mathcal{L}_{\text{swapper}}$; 4. the dependency regularized perturbation (denoted as DeepDRK[†]). Five synthetic datasets are considered as before and we report the values for FDR and Power under each setup. The results are presented in Figure 10.

Figure 10 illustrates the clear drawbacks of using only a single swapper ($K = 1$) and the case without the loss term REx. In those two cases, we fail to control the FDR on all datasets. The REx term is essential, as it ensures that the adversarial attacks generated by different swappers are addressed simultaneously. $\mathcal{L}_{\text{swapper}}$ is also an important term, as it promotes a diverse adversarial environments. Without this term we observe an increase in FDR compared to DeepDRK or DeepDRK[†]. The difference between DeepDRK and DeepDRK[†] is more subtle as DeepDRK[†] has already obtained high quality knockoffs. However, we empirically observe that adding the perturbation further increases the power for some datasets without sacrificing the FDR controllability. To interpret these observations, we follow a similar procedure in Section 4.1 to study the distribution of the knockoff statistics. Results are included in Section I.3.

Overall, we verify that all terms are necessary components to achieve higher powers and controlled FDRs through the ablation studies.

## I.3 Analysis on the distribution of knockoff statistics

We present the additional results on the means and standard deviations of both null and nonnull knockoff statistics under various experimental setups.

Figure 11 shows the results on $w_j$ for $n = 2000$ and it complements the results on Power and FDR in Figure 2. We notice that all models have concentrated null features around zero and relatively high values for the nonnull knockoff statistics for the case with 2000 samples. This corresponds to consistently accurate FS results across all models and datasets, as shown in Figure 2.

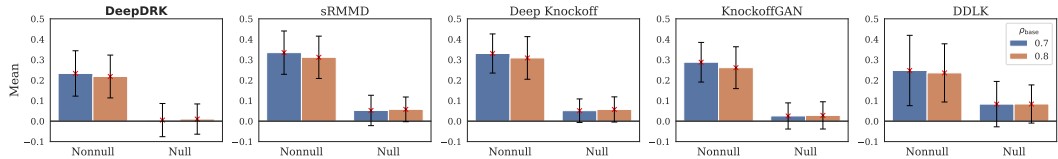

Figure 12: Knockoff statistics ($w_j$) for different models with the increased correlation of the Gaussian mixture data. Each bar in the plot represents the mean of the null/nonnull knockoff statistics averaging on 600 experiments. The error bar indicates the standard deviation. The sample size is 200.

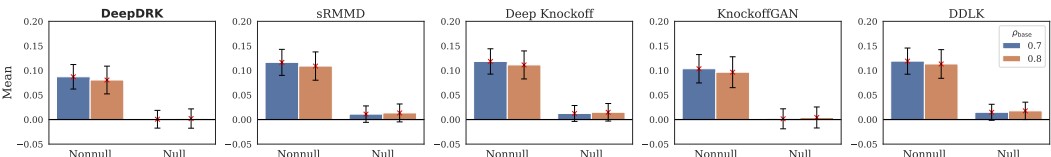

Figure 13: Knockoff statistics ($w_j$) for different models with the increased correlation of the mixture of Gaussians data. Each bar in the plot represents the mean of the null/nonnull knockoff statistics averaging on 600 experiments. The error bar indicates the standard deviation. The sample size considered is 2000.

Figure 12 and Figure 13 show the results for the Gaussian mixture data with increased correlations ($\rho_{\text{base}} = 0.7$ and $0.8$) for $n = 200$ and $n = 2000$ respectively. The figures complement the FDR-power results in Figure 3. It is clear for both $n = 200$ and $n = 2000$ cases, all models experience an increase in FDR and a decrease in power. This phenomenon can be reflected by the positive shifts of the knockoff statistics for the null features in Figure 12 and Figure 13. However, DeepDRK significantly controls the shifts, achieving the best results with the lowest FDRs and comparable power as shown in Figure 3.

In addition, we also compare the knockoff statistics for the models considered in the ablation studies in I.2. The results for both $n = 200$ and $n = 2000$ are in Figure 14 and Figure 15. Although the null knockoff statistics for DeepDRK, DeepDRK[†] and "No $\mathcal{L}_{\text{swapper}}$" models concentrate symmetrically around zero, the heights of the nonnull knockoff statistics for DeepDRK are the highest, resulting higher power. And because some nonnull knockoff statistics have small values for the DeepDRK[†] and "No $\mathcal{L}_{\text{swapper}}$" cases, we also expect them to have higher FDRs (see Figure 10). On the other hand, compared to DeepDRK, the models of "No REx" and "$K = 1$" experience clear positive shifts for the null knockoff statistics, leading to higher FDRs during FS (see Figure 10).

### I.4 Training time

We consider evaluating and comparing model training runtime in Table 8 with the (2000, 100) setup, as it is common in the existing literature. Although DeepDRK is not the fastest among the compared models, the time cost—7.35 minutes—is still short, especially when the performance is taken into account.

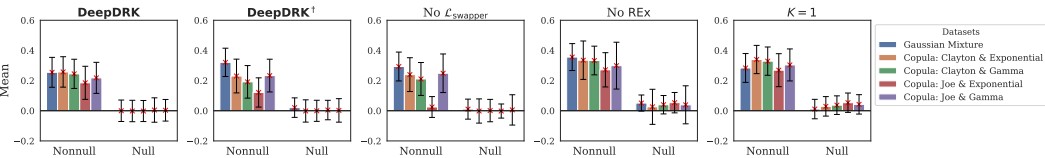

Figure 14: Knockoff statistics ($w_j$) for different models in ablation studies. Each bar in the plot represents the mean of the null/nonnull knockoff statistics averaging on 600 experiments. The error bar indicates the standard deviation. $K$ refers to the number of swappers. The sample size is 200.

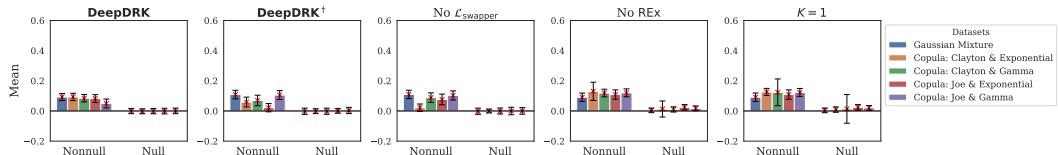

Figure 15: Knockoff statistics ($w_j$) for different models in ablation studies. Each bar in the plot represents the mean of the null/nonnull knockoff statistics averaging on 600 experiments. $K$ refers to the number of swappers. The sample size considered is 2000.

| DeepDRK | Deep Knockoff | sRMMD | KnockoffGAN | DDLK |
|---------|---------------|-------|-------------|------|
| 7.35 min | 1.08 min | 6.38 min | 10.52 min | 53.63 min |

Table 8: The training time for the models with $n = 2000$ and $p = 100$. The batch size is $64$ and the models are all trained with $100$ epochs.

## I.5  Additional Results for the IBD Study

Here we provide the supplementary information for the experimental results described in Section 4.3. In Table 9, we provide the list of identified metabolites by each of the considered models. This table provides additional information for Table 2 in the main paper which only includes metabolite counts due to limited space.

| Metabolite | **DeepDRK** | Deep Knockoff | sRMMD | KnockoffGAN | DDLK |
|---|---|---|---|---|---|
| 12.13-diHOME | | | | * | |
| 9.10-diHOME | | | | | |
| caproate | * | * | * | | * |
| hydrocinnamate | | | | | |
| mandelate | | | | | |
| 3-hydroxyoctanoate | | | | | |
| caprate | | | | | |
| indoleacetate | | | | | * |
| 3-hydroxydecanoate | | | | | |
| dodecanoate | | | | * | |
| undecanedionate | * | | | * | |
| myristoleate | | | | | |
| myristate | | | | | |
| dodecanedioate | | | | * | |
| pentadecanoate | | | | | |
| hydroxymyristate | | | | | |
| palmitoleate | | | | | |
| palmitate | | | | * | |
| tetradecanedioate | | * | | | |
| 10-heptadecenoate | | | | | |
| 2-hydroxyhexadecanoate | | | | | |
| alpha-linolenate | | | | | * |
| linoleate | | | | | |
| oleate | | | | | |
| stearate | | | | | * |
| hexadecanedioate | | * | | * | * |
| 10-nonadecenoate | | | | | |
| nonadecanoate | | | | | |
| 17-methylstearate | * | * | | | * |
| eicosapentaenoate | * | * | | | * |
| arachidonate | * | * | | * | * |
| eicosatrienoate | * | * | | | * |
| eicosadienoate | * | * | * | | * |
| eicosenoate | | | | | |
| arachidate | | | | * | |
| phytanate | | | | | |
| docosahexaenoate | * | * | | | * |
| docosapentaenoate | * | * | | * | * |
| adrenate | * | * | * | * | * |
| 13-docosenoate | | | | | |
| eicosanedioate | * | * | | | |
| oleanate | | | | | |
| masilinate | | | | | |
| lithocholate | * | | | | |
| chenodeoxycholate | | | | | |
| deoxycholate | * | | | * | * |
| hyodeoxycholate/ursodeoxycholate | | | | | |
| ketodeoxycholate | * | | | | |
| alpha-muricholate | * | | | | |
| cholate | | * | | | |
| glycolithocholate | * | | | | |
| glycochenodeoxycholate | | | | | |
| glycodeoxycholate | | | | | |
| glycoursodeoxycholate | | | | | |
| glycocholate | | | | | |
| taurolithocholate | | | | | * |
| taurochenodeoxycholate | | | | | |
| taurodeoxycholate | | | | | |
| taurohyodeoxycholate/tauroursodeoxycholate | | | | | |
| tauro-alpha-muricholate/tauro-beta-muricholate | | * | | | * |
| taurocholate | | | | | |
| salicylate | * | * | * | * | |
| saccharin | | | | * | |
| azelate | | | | | * |
| sebacate | * | | | | * |
| carboxyibuprofen | | | | | |
| olmesartan | | | | | |
| 1.2.3.4-tetrahydro-beta-carboline-1.3-dicarboxylate | | | | | |
| 4-hydroxystyrene | | * | | * | * |
| acetytyrosine | | | | | |
| alpha-CEHC | | | | | |
| carnosol | | | | | * |
| oxypurinol | | | | | |
| palmitoylethanolamide | | | | | |
| phenyllactate | * | * | * | | * |
| p-hydroxyphenylacetate | * | * | | | * |
| porphobilinogen | * | | | | |
| urobilin | * | * | | * | * |
| nervonic acid | | | | | |
| oxymetazoline | * | * | | | * |

Table 9: A list of literature-supported metabolites out of a total of 80 candidates. "*" indicates the important metabolites marked by the corresponding algorithms.

