# OpenReview forum: "DeepDRK: Deep Dependency Regularized Knockoff for Feature Selection"
_NeurIPS.cc/2024/Conference — NeurIPS 2024 poster_

### Official Review · Reviewer_FDZg · 2024-07-18

**Soundness:** 2
**Presentation:** 1
**Contribution:** 2
**Rating:** 3
**Confidence:** 3

**Summary:**

This paper proposes a deep-learning-based method for performing "knockoff"-based feature selection. The main contribution is the proposal of a new optimization problem, which incorporates a four part loss.  The method is evaluated on synthetic, semi-synthetic, and real-world data, showing comparable and, at times, improved performance over relevant baselines.

**Strengths:**

**Experiments.** At times, this paper shows that the proposed method improves the power for a fixed FDR.

**Weaknesses:**

**Motivation.** It's not clear why this problem is of interest. It is true that many papers have been published on this topic. But the authors motivate the problem not by describing why knockoffs are particularly important; they simply say that finding information features is "mission impossible," which doesn't give the reader any idea of why the problem is relevant.

**Organization.** To articulate a list of contributions, a paper must clearly answer two fundamental questions: (1) "What is the problem under consideration?" and (2) "How does the proposed method address this problem?" Unfortunately, it's not clear that this paper answers the first or second questions.  Here are some comments:
1. The authors claim that their method reduces reconstructability.  However, it's not clear what "reconstructability" is, or how it would be formally defined. The idea of reconstructability only arises in the discussion of related work, and even here, the problem is not presented in a way that is distinct from the solution presented in [4].
2. Section 2.1 is confusing. More care should be taken w/r/t the introduction of the problem of generating a knockoff sample.  The authors adopt quite a bit of notation from [11].  However, whereas [11] presents the method with examples and clearly articulated goals, this paper's summary is so coarse/brief that any reader who has not read this past work will have little to no chance of understanding the problem of finding knockoffs.
3. Deferring the definition of FDR to the appendix doesn't make sense to me. This quantity is of fundamental interest; it is measured throughout the experiments, and theorems are presented (in part from past work) which rely on this definition. The paper is incomplete without including this definition in the main text.
4. Theorem 2.1 is not clearly stated. The "aforementioned property" is not inferable from context.  The authors should make it clear that (1) they are referring to Theorem 1 in [11] and (2) that the "aforementioned property" is the choice of threshold in (3).
5. Readability would be improved if the authors included an example of the swap function, such as the example in [11] after Def. 2.
6. The authors seem to use (X,Z) and [X,Z] freely and interchangeably. This should be clarified. [11] sticks with (X,Z) (rather than square brackets) and I would recommend that the authors do the same.
7. Make it abundantly clear that whenever the authors say "power," they mean low type II error. This may not be clear to readers coming from deep learning.

**Main method.** A few comments on the main method.
1. The results are presented in a confusing order. (4) is presented before the authors know what any of the pieces are. Readers must wait around a page to get the details, which will frustrate. Furthermore, it's not clear why (4) is a fundamentally "good" objective, or where it came from. As written, it feels like the authors pulled this problem out of a hat.  Why does posing this as a minimax problem (as opposed to, e.g., a constrained problem) make sense? Why do the authors choose these two loss terms (and not others)? And how should one choose all of the trade-off parameters?
2. The authors seem to imply that [56] has a problem. It's not clear what this problem is; the "is far from enough" sentence doesn't give any quantitative evidence as far as I can tell. The result of this is that when the authors say "To fix it, ...," it's unclear what problem they are fixing.
3. The authors borrow a loss term from the risk extrapolation (REx) paper from domain generalization. The authors should justify this choice, as well as the other loss functions that they append onto their objective. As written, it feels as if the authors are somewhat arbitrarily adding on loss functions, and there isn't a clear principle driving the decision making. This gives the paper a largely heuristic feel. And while the authors claim to provide ablations in the appendix, it's not clear where they are. The authors say that the results are "in the associated tables," but it's not clear what tables these are. A similar concern arises w/r/t Figure 9.
4. The trade-off between L_SL and L_DRL isn't clearly presented. This content should be presented in the main text; based on the figure in Appendix G, it's not clear that the trade-off always holds. Furthermore, even if it did hold, the contribution that the authors are "the first to observe this phenomenon" seems to be trivially true: If no one has used this loss before, then of course no one has seen this trade-off.

**Experiments.** Some comments on the experiments.  The experiments seem to show marginal improvements for the proposed method. While DeepDRK does at times achieve higher power for a fixed FDR, the results are often fairly marginal, e.g., in Figure 5.  Given the shortcomings in the presentation of the method, it's not clear that these results do enough to justify acceptance.

**Questions:**

See above.

---

> ### Author Rebuttal · Authors · 2024-08-07
>
> Thank you for your comments. Please refer to the following for our response.
>
> Comment: it's not clear why (4) is fundamentally "good" …
> - Response: Thank you for the comments and questions. We will add more descriptions on the motivation for the two terms in the revised manuscript. Essentially, the two general terms are introduced to enforce swap property and to reduce collinearity, respectively. This form of losses are seen in most existing deep-learning-based knockoff methods [1][2][3][4]. Nonetheless, our proposed version, as empirically verified in all the experiments presented in the paper, suggests a good performance during the FDR-controlled feature selection. Adopting the min-max formulation is natural, as we want to minimize the total loss over the swapper that maximizes the distance between $(X, \tilde{X})$ and $(X, \tilde{X})_{\text{swap}(B)}$.
>    For the hyper parameters, we chose based on the performance across different dataset experiments and identified that the configuration outlined in Table 2 in general provides good controlled FDR and high power. All results (with different datasets) presented in the paper relies on this single setup, suggesting the adaptability of this method across different application scenarios.
> - [1] Romano, Y., et al. . Deep knockoffs. ` Journal of the American Statistical Association, 115(532):1861–1872, 2020.
> - [2] Jordon, J., et al. : Generating knockoffs for feature selection using generative adversarial networks. In International Conference on Learning Representations, 2018.
> - [3] Sudarshan, M., et al.. Deep direct likelihood knockoffs. Advances in neural information processing systems, 33:5036–5046, 2020.
> - [4] Masud, S. B., et al.. Multivariate rank via entropic optimal transport: sample efficiency and generative modeling. arXiv preprint arXiv:2111.00043, 2021.
>
> Comment: The authors seem to imply that [56] has a problem. It's not clear…
> - Response: Thanks for pointing out the ambiguity. [56] uses single-swapper during training, which results in insufficient FDR control and selection power (see Figure 2). The necessity of introducing the multi-swapper setup can be justified by the nonlinearity of learning the knockoff copies from X in the deep learning setup. Given that this optimization is nonlinear, there could be multiple local optima that the swap property could be violated. Introducing this multi-swapper scheme would enforce the model on learning a good knockoff that improves the feature selection performance. We have included the ablation study on the effectiveness of this multi-swapper setup. Please refer to Figure 9 for comparing between the DeepDRK model and the K=1 model. We can see that a single-swapper setup results in failure of FDR control.
>
> Comment: The authors borrow a loss term from the risk extrapolation (REx) …
> - Response: The first SWD term in Eq. (5) is introduced to directly enforce the swap property outlined in Eq. (1). There could be multiple choices such as KL divergence, Wasserstein Distance, etc. We choose SWD over other measurements due to the runtime benefit brought by the SWD (see Table 1 for the runtime comparison to other methods). The rationale behind the introduction of REx resides in the lack of FDR controllability in the single-swapper setup. Empirically in the case with only one swapper, we observe that the null knockoff statistics are not symmetric around zero, therefore it is hard to control the FDR and (see Figure 9 and Figure 13 in Appendix K2 and K3 for empirical results). This indicates that the swap property is not well-enforced. Thus we propose to use a multi-swapper setup. Each swapper represents one adversarial environment, so we are optimizing under multiple adversarial attacks. In this case, the REx term is naturally introduced to fight against multiple adversarial attacks simultaneously. Results can also be found in Figure 9 and Figure 13. The last $L\_\text{swapper}$ term is to ensure different adversarial environments. We have provided explanations of these terms in the paragraphs below Eq. (5) on page 4. We have improved the writing for better clarity in the revised paper.
>
> Comment: The trade-off between L_SL and L_DRL isn't clearly presented. This content…
> - Response: We empirically observe a competition between the two losses from the fact that when one loss decreases, the other increases, and it is hard to find a good set of hyper-parameters $[\lambda_1 , \lambda_2]$ to strike a balance. It is not a training problem because, as stated in Section 3.2, we observe such a phenomenon in all baseline knockoff generation algorithms of this paper. The baseline methods of course don’t define the losses in the exact same way as we did, but they all have corresponding loss terms that serve the same purposes. Thus the observation is well worth pointing out and the underlying reasoning is also stated in our paper: minimizing the swap loss, which corresponds to FDR control, is the same as controlling Type I error. Similarly, minimizing the dependency regularization loss is to control Type II error. With a fixed number of observations, it is well known that Type I error and Type II error can not decrease at the same time after reaching a certain threshold. This is why the two losses always compete with each other.
>
> Comment: The experiments seem to show marginal…
> - Response: Only looking at the right column in Figure 5, the improvement of our method is not significant, because the linear synthetic rule is an easy task, on which all methods perform quite well. However when a tanh synthetic rule is applied, i.e. the left column of Figure 5, it is apparent that DeepDRK is the only method that controls FDR under the target level and at the same time enjoys high selection power. Other results such as Figure 2 and 3 also suggest that for complex distributions and low sample size, DeepDRK performs significantly better than baseline algorithms.

---

> ### Author Response · Authors · 2024-08-07
>
> Thanks for your valuable comments. Due to page limit, we provide brief introductions to the standard knockoff definitions and the corresponding selection procedures, as the general motivation and idea have been widely known by the statistics community. We do understand that it could cause inconvenience to people unfamiliar with the subject, and hereby in the revised manuscript we have added more details. We have also reorganized the paper and fixed typos based on your advice.
>
> Feature selection, or variable selection, is a data-driven method to determine which of the $p$ features $(X\_1, \ldots, X\_p)$ are statistically related to the response $y$ in a linear model. Feature selection methods are important in a number of application areas, such as medical healthcare, economy, and political science [1]. Among various selection procedures, model-X knockoff emerges to be useful in providing an effective way to control the false discovery rate (FDR) of the selection result, regardless of the regression algorithm. To improve readability, we have already moved the definition of FDR to Section 2.1 from Appendix A and added examples to better illustrate swap property in the Appendix.
>
> The knockoff selection method proposed in most statistics literatures [1] has two limitations. First, the assumption that $X$ is Gaussian is often unrealistic, and its underlying distribution is usually unknown. Second, simply decorrelating $X\_j$ and $\tilde{X}\_j$ is not enough whenever $X\_j$ is almost surely equal to a linear combination of $X\_{-j}$ [2], which is stated in Appendix B. This is referred to as “reconstructability”, which is a population counterpart of collinearity, see Section 2.3 and Appendix B for more discussions. When there is reconstructability, the resulting knockoff adds collinearity into the regressor $(X,\tilde{X})$, deteriorating the regression and thus the selection result. Due to the two drawbacks, we propose DeepDRK, a framework that performs powerful knockoff-based feature selection for unknown $X$ distributions.
>
> - [1] Candes, E., Fan, Y., Janson, L., & Lv, J. (2018). Panning for gold:‘model-X’knockoffs for high dimensional controlled variable selection. Journal of the Royal Statistical Society Series B: Statistical Methodology, 80(3), 551-577.
> - [2] Spector, A., & Janson, L. (2022). Powerful knockoffs via minimizing reconstructability. The Annals of Statistics, 50(1), 252-276.

---

> > ### Author Response · Authors · 2024-08-11
> >
> > Dear reviewer,
> > Thanks a lot for your time and effort. As the discussion period will end soon, we would greatly appreciate your feedback on whether our rebuttal has adequately addressed your concerns. Please reach out if you have further comments. Thank again!

---

### Official Review · Reviewer_Znex · 2024-07-23

**Soundness:** 3
**Presentation:** 3
**Contribution:** 3
**Rating:** 6
**Confidence:** 4

**Summary:**

This paper considers the construction of knockoff features
in the variable selection framework of model-X knockoffs.
The authors propose a deep learning-based method for
generating knockoff features. Extensive numerical simulation and real-data examples show
that the proposed method has advantageous performance compared
with other state-of-art methods, especially when the covariate distribution
is complex and when the sample size is relatively small.

**Strengths:**

The paper considers one fundamental problem in the model-X knockoff
pipeline --- the construction of valid and high-quality knockoffs.
This is key to the applicability of the knockoffs method.


Substantial engineering efforts have been made in this work and
the proposed method is demonstrated to have satisfactory performance
in extensive numerical experiments. This is a significant contribution to the
knockoffs literature.

**Weaknesses:**

The proposed method is based mostly on empirics and heuristics,
with a theoretical understanding lacking.

**Questions:**

1. The introduction of the dependency regularized perturbation is
for power boosting, and yet in the ablation study, this feature seems to help
more with FDR control as opposed to power-boosting. Is there a reason?
2. On page 4, line 132, the definition of REx: I am confused since
the LHS does not depend on $i$ but the RHS depends on $i$.
3. On page 4, line 148: "correltaion" -> "correlation"
4. On page 5, line 153: I am confused by the statement that "the knockoff
 $X_j$ should be independently sampled from $p_j(\cdot \mid X_{-j})$"
--- this alone is not enough to ensure the knockoff property.
5.  On page 5, line 173, what is $\alpha$?

**Limitations:**

The authors have adequately addressed the limitations.

---

> ### Author Rebuttal · Authors · 2024-08-07
>
> Thank you for your comments and advice. Please checkout our responses below.
>
> Question: The introduction of the dependency regularized perturbation is for power boosting, and yet in the ablation study, this feature seems to help more with FDR control as opposed to power-boosting. Is there a reason?
> - Response: Thank you for your question. In principle DRP should be considered as an approach to reduce collinearity in the concatenated design matrix at the cost of increasing the swap loss. It is true that the design of the DRP technique does not aim for FDR reduction. Nonetheless, looking at the knockoff statistics $W\_j$ in Figure 4 and Figure 13, we clearly observe that DRP tends to make the null statistics symmetric around zero. Since the knockoff statistics is the variable that is used for feature selection, the symmetric observation will reduce the number of negative $W\_j$’s to be selected, according to the selection criteria Eq. (3). This leads to a lower FDR.
>
>
> Question: On page 4, line 132, the definition of REx: I am confused since the LHS does not depend on i but the RHS depends on i
> - Response: Thank you for pointing this out. We will revise the equation to $\text{REx}(X, \tilde{X}\_{\theta}, \\{S\_{\omega\_i}\\}_{i=1}^K)  = \widehat{\text{Var}}\_{S\_{\omega}}(\text{SWD}([X, \tilde{X}\_\theta], [X, \tilde{X}\_\theta]\_{S\_{\omega_i}}); i \in [K])$ to make the LHS and RHS consistent.
>
> Question: On page 4, line 148: "correltaion" -> "correlation"
> - Response: Thank you for pointing this out, we will make corrections in the revision.
>
> Question: On page 5, line 153: I am confused by the statement that "the knockoff $\tilde{X}\_j$ should be independently sampled from $p(\cdot \vert X\_{-j})$ --- this alone is not enough to ensure the knockoff property.
> - Response: Thanks for pointing out. Indeed the description is a bit hand-waving. To be precise, we want to sample $\tilde{X}\_j$ from $p\_j(\cdot | X\_{-j})$ such that $X\_j$ and $\tilde{X}\_j$ are as less dependent as possible. This ensures the swap property but reduces reconstructability. We will revise the description in the manuscript.
>
> Question: On page 5, line 173, what is $\alpha$
> - Response: Thank you for pointing this out. This is a typo and it should be $\alpha\_n$. We will correct it in the revision.

---

> > ### Author Response · Authors · 2024-08-11
> >
> > Dear reviewer,
> > Thanks a lot for your time and effort. As the discussion period will end soon, we would greatly appreciate your feedback on whether our rebuttal has adequately addressed your concerns. Please reach out if you have further comments. Thank again!

---

> > > ### Comment · Reviewer_Znex · 2024-08-12
> > >
> > > I want to thank the authors for their response. I find my concerns addressed, and remain positive about this submission.

---

### Official Review · Reviewer_WTXG · 2024-07-29

**Soundness:** 4
**Presentation:** 4
**Contribution:** 3
**Rating:** 7
**Confidence:** 4

**Summary:**

The work introduces DeepDRK, a new algorithm for improving FDR in the model-X knockoff framework. In this framework, a knockoff covariate is generated for each existing covariate where then knockoff covariates have to satisfy the swap property. Given a knockoff statistic that satisfies the flip-sign property, one can perform feature selection with guaranteed false discovery rate by training a regression/classification model on the concatenated vector of original and knockoff covariates and then using any arbitrary feature importance score. Therefore, the main challenge in this framework is generating the knockoffs such that they satisfy the strong swap property: swapping any arbitrary subset of covariates with their knockoffs should not change the distribution of the concatenated vector.

Given that most real-world data does not follow Gaussian or a mixture of Gaussian distribution, generating knockoffs that satisfy the swap property is a challenge. Existing work is focused on using deep generative models (e.g. GANs) to handle this challenge. The paper argues that this approach for continuous covariates will result in reconstructability: overfitting on the data will make the distribution of knockoffs too similar to the data distribution and therefore making the concatenated vector too multicollinear for any feature-importance method to have discovery power.

The DeepDRK framework seeks to tackle reconstructability by training the deep model in such a way that not only satisfies the swap property but also reduces reconstructability. This is achieved in two steps:
Firstly, by having a 1- the adversarial swap loss where a group of models compete adversarially against the knock-off generator model 2- a dependence regularization loss to reduce reconstructability by directly reducing the sliced-Wasserstein correlation between the original and knockoff vectors.
Secondly, by the post-training dependency regularization which interpolates between the generated knockoff data table and the row permuted original data table. By making the interpolation weaker or stronger, one gains control in how much they can improve how much swap condition holds and therefore how good the FDR is at the cost of loosing power and vice versa.
The paper then shows experimental evidence in synthetic and semi-synthetic scenarios (where the true covariate-response) relationship is known. The argument is that compared to existing algorithms that have better power in low-sample settings, DeepDRK's empirical FDR is close the the selected level more consistently and this is proven both by the FDR vs Power results and the original vs knockoff feature importance distributions.

**Strengths:**

All in all, the main novelties in the paper are:
- main one is the post-training Dependency Regularization Perturbation
- using multi swappers and SWD criterion
- Experimenting on synthetic distributions other than GMMs
- Showing that unlike other deep learning based knockoff algorithms, DeepDRK has more consistent FDR preservation. The ablation study really helps with understanding the benefit of each of the algorithmic choices.

**Weaknesses:**

- It seems necessary to see 4.1 results for various beta scales used for generating the response variable to compare different methods in various levels of difficulties to make sure the value of 15 has not been selected to favor DeepDRK. I think it'd be interesting to have the results in a 2D FDR vs Power scatter plot for better comparison.
- I'd be interested in seeing mean + confidence interval performance plots for Figure 2 and Figure 3 using various distribution parameter selections (e.g. for GMM distribution using various \pi parameters in addition to to more \rho_base, etc)
- The process for choosing the value of alpha=0.5 needs to be clarified as "it leads to consistent performance" might suggest choice based on final results which would be a case of overfitting. The discussion related to Figure 8 seems to confirm overfitting.

**Questions:**

Discussed above

**Limitations:**

There needs to be a discussion regarding the choice of hyperparameters especially the alpha parameter as in real-world applications of knockoffs it's impossible to know the best performing value in advance. Given the results in Figure 8, there seems to be high sensitivity to the alpha value which would make the framework unreliable. At the very least, the authors need to show the superiority of their framework to existing methods in a large interval of alpha values.

---

> ### Author Rebuttal · Authors · 2024-08-07
>
> Thank you for your comments and advice. Please checkout our responses below.
>
> Comment: It seems necessary to see 4.1 results for various beta scales used for generating the response variable to compare different methods in various levels of difficulties to make sure the value of 15 has not been selected to favor DeepDRK. I think it'd be interesting to have the results in a 2D FDR vs Power scatter plot for better comparison.
> - Response: Thank you for the suggestions. We have provided experimental results concerning 4 different sets of the scale value: 5, 10, 15, 20. Results are presented in Figure 1 of the attached PDF file in the Author Rebuttal section. From the figure, we can clearly observe that the proposed DeepDRK outperforms other methods for a controlled FDR with relatively higher power (e.g. points are closer to the top left region). In addition, this figure also conveys a message that the commonly used $\frac{p}{\sqrt{n}}$ (with n = 2000) setup in existing works is a relatively easy task where most methods can achieve controlled FDR with high power.
>
> Comment: I'd be interested in seeing mean + confidence interval performance plots for Figure 2 and Figure 3 using various distribution parameter selections (e.g. for GMM distribution using various $\pi$ parameters in addition to to more $\rho_\text{base}$, etc)
> - Response: Thank you for the comments. We have provided GMM distribution with 10 sets of different $\pi$ parameters. Results are presented in Figure 2 and Table 1 of the attached PDF. This should complement Figure 2 and 3 of the main manuscript on the change of $\rho_base$ for the GMM distributions. From Figure 2, we visually observe that the model performance does not vary significantly for all tested models on 3 out of 10 $\pi$ setups. Among all the models, the proposed DeepDRK is the only one that can control the FDR given the nominal 0.1 level. In addition, we also provide a table that includes the mean, standard deviation and median, 5\% and 95\% quantiles for the 10 complete setups. This should reveal the robustness of the proposed model in this setup.
>
> Comment: The process for choosing the value of alpha=0.5 needs to be clarified as "it leads to consistent performance" might suggest choice based on final results which would be a case of overfitting. The discussion related to Figure 8 seems to confirm overfitting.
> - Response: First of all, please allow us to make corrections on Eq. (8). We realized that Eq. (8) has a missing coefficient $1- \alpha\_n$ in front of  $\tilde{X}\_{\theta}$ . Essentially we perform interpolation between $\tilde{X}\_\theta$ and $X\_\text{rp}$ as the former seeks to control FDR for a satisfied swap property while the latter seeks to reduce collinearity for higher power. We look for a balance between the two. The corrected equation is $\tilde{X}^{\text{DRP}}\_{\theta, n} = (1 - \alpha\_n) \cdot \tilde{X}\_\theta + \alpha\_n \cdot X\_{\text{rp}}$ and this will be reflected in the revised manuscript. In addition, Figure 8 was plotted according to an earlier version of Eq. (8), where we take $\tilde{X}^{\text{DRP}}\_{\theta, n} =  \alpha\_n \cdot \tilde{X}\_\theta + (1-\alpha\_n) \cdot X\_{\text{rp}}$.  In other words, the $\alpha\_n$ presented in Figure 8 refers to the coefficient for $\tilde{X}_\theta$, whereas $\alpha_n$ should refer to the coefficient for $X\_\text{rp}$ in the updated Eq. (8). We have corrected Figure 8 and present the new figure as Figure 3 in the attached PDF.
>
>     As for the consistency of the model on the choice of alpha, we want to first point out that the choice of knockoff is not unique [1][2] and there is no closed form solution to the knockoff variable in non-Gaussian setups. As a result, it is hard to find a value for $\alpha_n$ from the data via procedures like cross-validation. We want to point out that the choice of $\alpha_n$ is not a result of overfitting. According to Figure 8 (i.e. Figure 3 in the updated PDF in the Author Rebuttal section), we clearly observed that choices of $\alpha_n$ around 0.5 results in low FDR with relatively high power across multiple datasets, suggesting its generalizability for a range of different data setups.
>
> - [1] Candes, E., Fan, Y., Janson, L., & Lv, J. (2018). Panning for gold:‘model-X’knockoffs for high dimensional controlled variable selection. Journal of the Royal Statistical Society Series B: Statistical Methodology, 80(3), 551-577.
> - [2] Spector, A., & Janson, L. (2022). Powerful knockoffs via minimizing reconstructability. The Annals of Statistics, 50(1), 252-276.

---

> > ### Author Response · Authors · 2024-08-11
> >
> > Dear reviewer,
> > Thanks a lot for your time and effort. As the discussion period will end soon, we would greatly appreciate your feedback on whether our rebuttal has adequately addressed your concerns. Please reach out if you have further comments. Thank again!

---

> > ### Comment · Reviewer_WTXG · 2024-08-12
> > **Response**
> >
> > I want to thank the authors for addressing all of my questions comprehensively. The empirical results now seem robust and reliable. I have changed my review scores accordingly.

---

### Author Rebuttal · Authors · 2024-08-07

Dear Reviewers,

We have completed more experiments, as suggested in your comments. Please refer to the PDF file for details.

Specifically, Figure 1 refers to additional experiments on the change of $\beta$ coefficient scales. We include three scales, 5, 10, and 20, in addition to the default 15. Namely, we consider $\frac{p}{5\cdot \sqrt{n}}$, $\frac{p}{10\cdot \sqrt{n}}$, $\frac{p}{15\cdot \sqrt{n}}$ and $\frac{p}{20\cdot \sqrt{n}}$ in total.

Figure 2 provides barplots for three different setups for the weights (i.e. $\pi_1$, $\pi_2$ and $\pi_3$) of the components in mixture Gaussian . They are: 1. [$\frac{1}{3}$, $\frac{1}{3}$, $\frac{1}{3}$]; 2. [0.2, 0.3, 0.5]; 3. [0.5, 0.3, 0.2]. The barplots compare the model performance on these three setups on FDR and Power. In addition, we further consider 7 other setups and calculate statistics on the FDR and power (e.g. mean, standard deviation, median and quantiles). The weights are uniformly sampled. The results can be found in Table1. Weights detail is presented below:
- [0.56215729, 0.38386235, 0.05398036]
- [0.42981112, 0.16832496, 0.40186392]
- [0.31650882, 0.32417065, 0.35932052]
- [0.31627876, 0.38820664, 0.2955146]
- [0.43860741, 0.48803919, 0.0733534]
- [0.31395573, 0.04100264, 0.64504163]
- [0.65634908, 0.28201858, 0.06163234]

We also provide the corrected Figure 8 (of the manuscript) in Figure 3 in the PDF file. Essentially, it flipped the X-axis horizontally and nothing else was changed.

Sincerely,

The Authors of the Paper

---

### Decision · Program_Chairs · 2024-09-25

**Decision:**

Accept (poster)

**Comment:**

The paper introduces DeepDRK, a new algorithm for improving FDR (by to constructing knockoff features) in the model-X knockoff framework. The authors propose a deep learning-based method for generating knockoff features in which a knockoff covariate is generated for each existing covariate (and knockoff covariates have to satisfy the so-called swap property). The authors provide extensive numerical simulation and real-data examples to show that the proposed method has advantageous performance compared with other state-of-the-art methods, especially when the covariate distribution is complex and when the sample size is small.

All the reviewers agree that the paper considers an important question in the knockoff framework and gives an empirically satisfactory solution. Reviewers had some concerns about the presentation of the paper (e.g. additional background, motivation of the problem, etc) which have been resolved after the discussion period, and hence I would like to recommend the paper for acceptance. Also, I would like to suggest to the authors to revise the manuscript to improve the presentation according to the comments provided by the reviewers.